Brief Communication

# Droplet-based single-cell joint profiling of histone modifications and transcriptomes

Yang Xie[1,2,7], Chenxu Zhu ⬤[3,4,5,7], Zhaoning Wang[1], Melodi Tastemel[1], Lei Chang[1,3], Yang Eric Li ⬤[1,3] & Bing Ren ⬤[1,3,6] ✉

We previously reported Paired-Tag, a combinatorial indexing-based method that can simultaneously map histone modifications and gene expression at single-cell resolution at scale. However, the lengthy procedure of Paired-Tag has hindered its general adoption in the community. To address this bottleneck, we developed a droplet-based Paired-Tag protocol that is faster and more accessible than the previous method. Using cultured mammalian cells and primary brain tissues, we demonstrate its superior performance at identifying candidate *cis*-regulatory elements and associating their dynamic chromatin state to target gene expression in each constituent cell type in a complex tissue.

Chemical modifications to histone proteins and nucleic acids along chromosomes, collectively referred to as the cell's epigenome, regulate spatiotemporal gene expression patterns in multicellular eukaryotic organisms[1]. Analysis of the epigenome has been successfully used to reveal mechanisms of gene regulation and dysregulation in development, disease pathogenesis and aging; however, considerable technical barriers exist in profiling the epigenome of complex tissues due to the heterogeneity of cell types and states within biospecimens. Single-cell epigenomic techniques can circumvent this barrier by revealing the landscape of DNA methylation[2], chromosome conformation[3], chromatin accessibility[4], histone modifications[5–7] and transcription factor binding[8] at single-cell resolution. Single-cell multiomic assays that jointly survey multiple molecular modalities, including gene expression[9–13] or protein abundance[14,15], further help to decipher the complex interplay between the epigenome and transcriptional machinery. In particular, single-cell CUT&Tag (scCUT&Tag) has enabled the characterization of active or silenced chromatin states at candidate *cis*-regulatory elements (cCREs) within different cell types of primary tissues[16,17]. However, current single-cell epigenomic assays have been slow for general adoption, owing to factors such as lengthy procedures and lack of general accessibility.

Here, we report Droplet Paired-Tag, a fast and broadly accessible technique for producing high-quality single-cell joint profiles of histone modifications and transcriptomes in parallel, which has

the potential to be quickly adopted by the research community for interrogating the dynamic and cell-type-specific epigenomic landscapes in complex tissues. The key modifications compared to the initial combinatorial indexing-based Paired-Tag protocol include the adaptation of a commercially available microfluidic platform (that is, 10x Chromium Single Cell Multiome) to introduce cellular barcodes and a simplified protocol to prepare sequencing libraries. The new procedure, therefore, offers three key advantages. First, changing to a droplet-based platform greatly shortens the hands-on time in both the molecular barcoding and library preparation steps (Extended Data Fig. 1). As a result, Droplet Paired-Tag can be performed in less than 1.5 days from nuclei preparation to sequencing library construction, considerably shorter than the conventional 3-day-long Paired-Tag procedure. Second, this design can also be more easily adapted owing to the wide availability of the commercial 10x Chromium platform and reagent kits. Third, the simplified procedure also brings improved performance in identifying cCREs and correlating chromatin states of distal elements to expression levels of putative target genes.

In Droplet Paired-Tag, nuclei are first permeabilized, followed by targeted tagmentation using primary antibodies to a histone modification; these antibodies are precoupled with protein A–Tn5 transposase fusion proteins (pA–Tn5) in a procedure modified from the reported CUT&Tag[18] method (Methods). The resulting nuclei and barcoded beads are then coencapsulated into droplets with a microfluidic device

[1]Department of Cellular and Molecular Medicine, University of California, San Diego, San Diego, CA, USA. [2]Biomedical Sciences Graduate Program, University of California, San Diego, La Jolla, CA, USA. [3]Ludwig Institute for Cancer Research, La Jolla, CA, USA. [4]New York Genome Center, New York, NY, USA. [5]Department of Physiology and Biophysics, Institute for Computational Biomedicine, Weill Cornell Medicine, New York, NY, USA. [6]Center for Epigenomics, Institute of Genomic Medicine, Moores Cancer Center, University of California, San Diego, School of Medicine, La Jolla, CA, USA. [7]These authors contributed equally: Yang Xie, Chenxu Zhu. ✉e-mail: biren@health.ucsd.edu

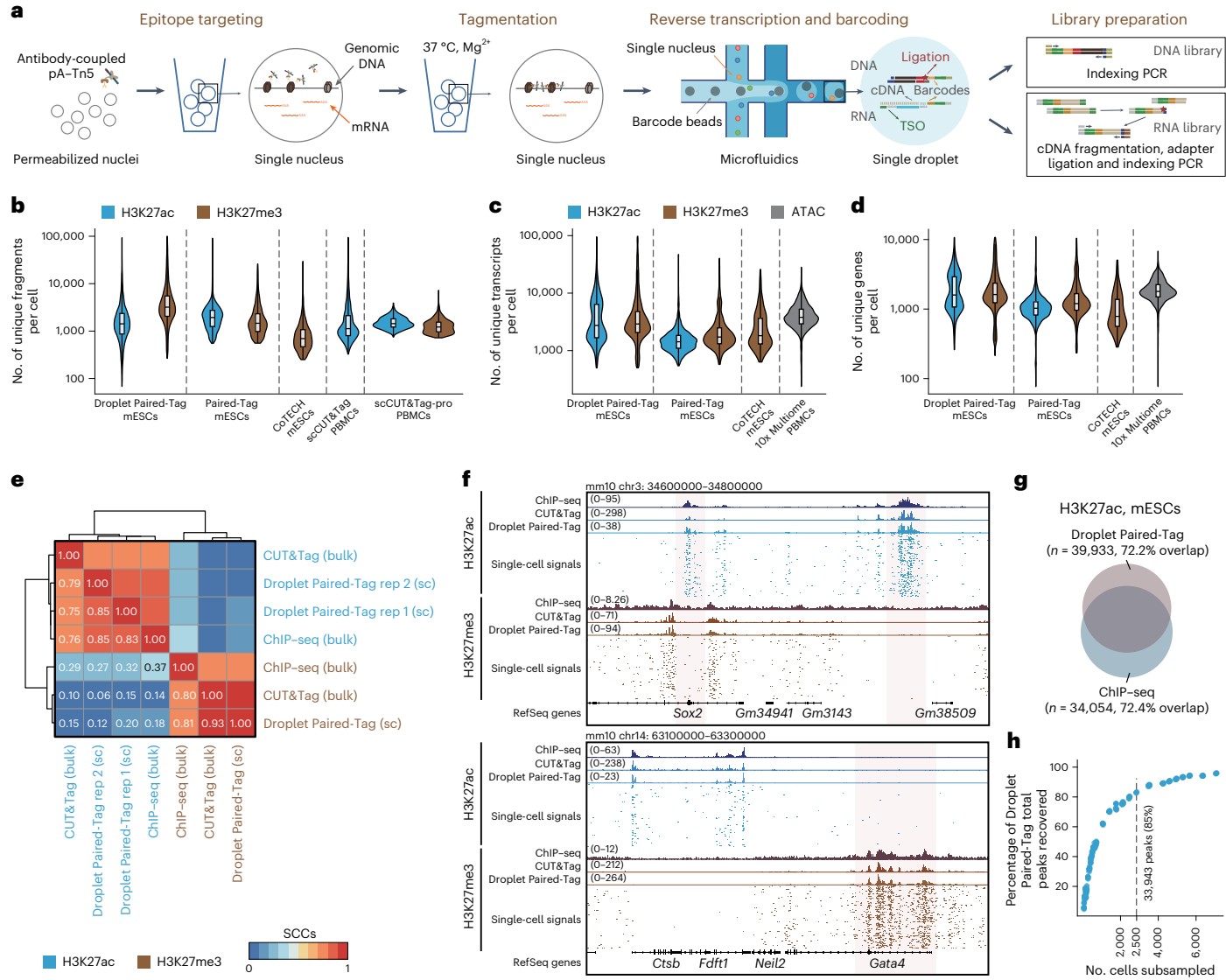

**Fig. 1 | Joint profiling of transcriptomes and histone modifications in single cells using Droplet Paired-Tag. a**, Schematic overview of Droplet Paired-Tag; TSO, template switch oligo. **b**, Distribution of the unique number of fragments per cell across different single-cell epigenomic assays with the indicated samples and histone targets; $n$ = 13,664 (Droplet Paired-Tag, H3K27ac), 4,501 (Droplet Paired-Tag, H3K27me3), 2,066 (Paired-Tag, H3K27ac), 418 (Paired-Tag, H3K27me3), 3,031 (CoTECH, H3K27me3), 5,111 (scCUT&Tag, H3K27ac), 9,039 (scCUT&Tag-pro, H3K27ac) and 12,585 (scCUT&Tag-pro, H3K27me3); PBMCs, peripheral blood mononuclear cells. **c,d**, Distribution of unique transcript (**c**) or gene (**d**) numbers per cell across different single-cell multiomic assays. For all box plots, hinges were drawn from the 25th to 75th percentiles, with the middle line denoting the median and whiskers denoting a maximum 2× the interquartile range; $n$ = 13,664 (Droplet Paired-Tag, H3K27ac), 4,501 (Droplet Paired-Tag,

H3K27me3), 2,066 (Paired-Tag, H3K27ac), 418 (Paired-Tag, H3K27me3), 3,031 (CoTECH, H3K27me3) and 2,711 (10x Multiome, ATAC). **e**, Genome-wide SCCs between mESC histone modification datasets from Droplet Paired-Tag (indicated with 'sc'), bulk CUT&Tag and ChIP–seq (indicated with 'bulk'); rep, replicate. **f**, Genome browser view showing examples of pseudo-bulk and single-cell histone modification signals of Droplet Paired-Tag in mESCs, along with bulk ChIP–seq and CUT&Tag. The pluripotent gene (*Sox2*), along with its superenhancer (SCR), and the repressed gene (*Gata4*) are highlighted in pink; chr, chromosome. **g**, Number and overlap of peaks called using H3K27ac Droplet Paired-Tag compared to those from ChIP–seq datasets. **h**, Scatter plot showing the relationships between the total number of H3K27ac peaks called and the number of nuclei after downsampling. The dashed line indicates the cutoff of the number of nuclei that could recover 85% of peaks.

(10x Genomics Chromium X controller). Two types of oligonucleotides with the same set of barcodes are embedded in the beads: (1) a barcoded poly(dT) oligonucleotide to label cDNA and (2) a capturing oligonucleotide to label DNA fragments derived from tagmentation. Reverse transcription is performed in the droplets with the barcoded poly(dT) oligonucleotide. A ligation reaction is simultaneously performed to attach barcoded capturing oligonucleotide to the tagmented chromatin fragments. To facilitate this ligation process with our in-house pA–Tn5 fusion, we designed the pA–Tn5 adaptor with 3'-extended bridge-pMENTs sequence reverse complementary to the capturing

oligonucleotide sequence (Fig. 1a, Extended Data Fig. 2a and Supplementary Table 1). After completion of the reactions, the reverse transcription products and tagmented DNA fragments from the same cells are both labeled with the same unique cellular barcodes. The droplets are then dissolved, and the cDNA and chromatin fragments are purified, amplified and split for sequencing library construction following the manufacturer-recommended protocols (10x Chromium Single Cell Multiome; Extended Data Fig. 2b–d).

As a proof of concept, we first used Droplet Paired-Tag to analyze histone modifications (H3K27ac and H3K27me3) jointly with the

transcriptome in individual mouse embryonic stem cells (mESCs). The complexity of the DNA-dedicated libraries corresponding to histone modifications (median number of 1,448 and 3,224 fragments per nucleus for H3K27ac and H3K27me3, respectively) was comparable to that of the combinatorial indexing-based Paired-Tag. Library complexity was also similar to or higher than library complexities from other published methods for analyzing histone modifications in single cells, such as scCUT&Tag[7], CoTECH and scCUT&Tag-pro (Fig. 1b). Compared to H3K27ac, a higher fraction of reads from H3K27me3 Droplet Paired-Tag experiments correspond to mono-, di- and trinucleosome fragments (Extended Data Fig. 3b), consistent with the more compact chromatin structures within Polycomb complex-repressed regions[19]. For the transcriptomic analysis, Droplet Paired-Tag yielded gene expression-dedicated libraries with comparable complexities as other commonly used single-cell RNA-sequencing (scRNA-seq) platforms (Fig. 1c,d). The gene expression profile detected by Droplet Paired-Tag was in excellent agreement with bulk RNA-seq data from mESCs (Extended Data Fig. 3c). As expected, the detected gene expression levels are, in general, positively correlated with the H3K27ac signal over the transcription start site (TSS) and inversely correlated with H3K27me3 deposition across gene bodies (Extended Data Fig. 3d). These results indicate that Droplet Paired-Tag can reliably capture histone modifications and gene expression simultaneously from the same cell in a high-throughput fashion.

To evaluate the sensitivity and specificity of Droplet Paired-Tag in histone modification profiling, we compared the mESC single-cell data with bulk CUT&Tag and public chromatin immunoprecipitation with sequencing (ChIP–seq) datasets generated from mESCs[20,21]. For both H3K27ac and H3K27me3, the aggregated single-cell signals faithfully resembled those from bulk CUT&Tag and ChIP–seq experiments (Fig. 1e) and showed high enrichment over peaks identified from ChIP–seq datasets (Extended Data Fig. 3e). As an example, single-cell H3K27ac reads from Droplet Paired-Tag (Fig. 1f) marked the promoter region of the pluripotent gene *Sox2* and its downstream superenhancer, while H3K27me3 reads were deposited on genes involved in cellular differentiation and genes that are expressed in specific cell lineages (for example, *Gata4*). Seventy-two percent of the peaks identified from aggregated single-cell signals from Droplet Paired-Tag overlapped with those from ChIP–seq experiments (Fig. 1g). By downsampling the number of nuclei profiled in Droplet Paired-Tag experiments, we found that the number of H3K27ac peaks detected reached saturation after about 2,500 nuclei, or around 6 million total reads. By comparison, the Paired-Tag required 60% more reads to reach saturation (Fig. 1h and Extended Data Fig. 3f).

To demonstrate the utility of Droplet Paired-Tag for single-cell epigenomic analysis of primary tissues, we used it to analyze histone modifications and gene expression at single-cell resolution in the adult mouse frontal cortex. We performed Droplet Paired-Tag experiments to profile either H3K27ac or H3K27me3 together with RNA expression, each in three biological replicates. After filtering out low-quality cells and potential doublets, we recovered 22,054 nuclei in total, of which 11,874 nuclei were profiled for H3K27ac and 10,180 were profiled for H3K27me3. Clustering of these nuclei based on their transcriptomic profiles identified 20 major cell clusters corresponding to nine glutamatergic neuron types (*Snap25*+*Slc17a7*+), six GABAergic neuron types (*Snap25*+*Gad1*+) and five non-neuron cell types (*Snap25*−; Fig. 2a, Extended Data Fig. 4a–d and Supplementary Tables 2 and 3). Most cell types were known to exist in the frontal cortex regions, except for three cell types found in one sample, namely, D12MSN (striatum D1/D2 like medium spiny neurons), OBGA (olfactory bulb GABAergic neurons) and OBGL (anterior olfactory nucleus glutamatergic neurons), which likely originate from anatomical regions proximal to the frontal cortex due to variations in surgical sectioning during sample preparation (Extended Data Fig. 4b and Supplementary Table 4). Interestingly, 17 out of 20 and 18 out of 20 clusters can be independently recovered by clustering using histone modification H3K27ac or H3K27me3 signals (Fig. 2a), respectively, suggesting that cell-type annotations are highly concordant between transcriptome- and epigenome-based clustering (Fig. 2b and Extended Data Fig. 5a–d). Additionally, the cell clusters reported in this study were also consistent with those from the previous Paired-Tag dataset and the BRAIN Initiative Cell Census Network (BICCN) reference 10x single-nucleus RNA-seq (snRNA-seq) dataset generated from the mouse primary motor cortex (Extended Data Fig. 5e–h)[22].

To jointly analyze Droplet Paired-Tag data corresponding to different histone modifications, we used transcriptome-based clustering and cell-type annotation in all subsequent analyses. For histone modality, pseudo-bulk-level signals showed high concordance with both bulk CUT&Tag and ChIP–seq experiments (Extended Data Fig. 6a,b). Pseudo-bulk single-cell histone signals from cells within each cluster showed that the H3K27ac signal is abundant at TSSs of cell-type-specific genes, whereas these regions are generally silenced by H3K27me3 in other cell types (Fig. 2c and Extended Data Figs. 6c and 7a–d). Compared to scCUT&Tag[6], Droplet Paired-Tag yielded a comparable fraction of reads in peaks (FRiP) but higher numbers of unique fragments per nucleus for both histone modifications. Compared to combinatorial indexing-based Paired-Tag[12], Droplet Paired-Tag recovered fewer unique reads per nucleus but showed higher FRiP and thus captured a higher number of peak-associated reads. The improvements in both

**Fig. 2 | Droplet Paired-Tag effectively resolves multiple cell types in the mouse frontal cortex (FC) and identifies the cCREs within each cell type. a**, Uniform manifold approximation and projection (UMAP) embedding visualization of frontal cortex Droplet Paired-Tag data clustered and annotated based on the transcriptome (gene expression) and histone modifications (H3K27ac and H3K27me3). **b**, Overlap of shared annotations between transcriptome and epigenome clustering. **c**, Representative genome browser view of gene expression and H3K27ac and H3K27me3 distribution over cell-type-specific marker genes. **d**, Comparison of the number of unique transcripts and genes detected in each cell between Droplet Paired-Tag and Paired-Tag; *n* = 22,054 (Droplet Paired-Tag, RNA) and 11,026 (Paired-Tag, RNA). **e**,**f**, Comparison of the unique fragments and FRiPs in each cell between Droplet Paired-Tag and other methods measuring the histone modifications H3K27ac (**e**) and H3K27me3 (**f**) in single cells. All box plot hinges were drawn from the 25th to 75th percentiles, with the middle line denoting the median and whiskers denoting 2× the interquartile range; *n* = 11,874 (Droplet Paired-Tag, H3K27ac), 5,049 (scCUT&Tag, H3K27ac), 886 (Paired-Tag, H3K27ac), 10,180 (Droplet Paired-Tag, H3K27me3), 4,019 (scCUT&Tag, H3K27me3) and 60 (Paired-Tag, H3K27me3). **g**, Signal enrichment over CEMBA cCREs in Droplet Paired-Tag and other methods measuring single-cell histone modifications; kbp, kilobase pairs. **h**, Comparison of the number of H3K27ac peaks from the Droplet Paired-Tag dataset with the original Paired-Tag dataset, intersected with CEMBA-identified cCREs. **i**, Heat map showing gene expression values from promoter-proximal cCREs. **j**, Signals of both histone modifications over promoter-proximal cCREs across different cell types. **k**, Heat map of known motif enrichment for each cCRE module of promoter-proximal cCREs. Examples of known motifs are shown along with the heat map. *P* values were calculated by one-sided binomial test. FDRs were then calculated to select enriched motifs; TF, transcription factor. **l**, Schematics for identifying potential targets for cCREs. **m**, Frequency density plots showing the distribution of SCCs between distal cCRE histone modification signals and their putative target gene expression level. Cutoffs (FDR = 0.05) used to identify cell-type-specific cCRE–gene pairs are also indicated. **n**, Heat map showing histone modification signals at distal cCREs with potential active or repressive roles and their putative target gene expression levels. Example overrepresented GO terms for genes in selected cell types are shown. *P* values were calculated by Fisher's exact test. Benjamini–Hochberg FDRs were then calculated to select overrepresented GO terms.

signal sensitivity and specificity likely contributed to the higher resolution in separating cell types (Fig. 2d–f and Extended Data Fig. 6f–i). Compared to the list of open chromatin regions identified from the same brain cell types from a recent single-nucleus ATAC-seq atlas (BICCN)[23],

Droplet Paired-Tag yielded the lowest level of H3K27me3 signals at the open chromatin regions, indicating minimal off-target Tn5 transposase activities in our procedure (Fig. 2g). To evaluate the sensitivity of Droplet Paired-Tag, we identified the peaks of H3K27ac signals in

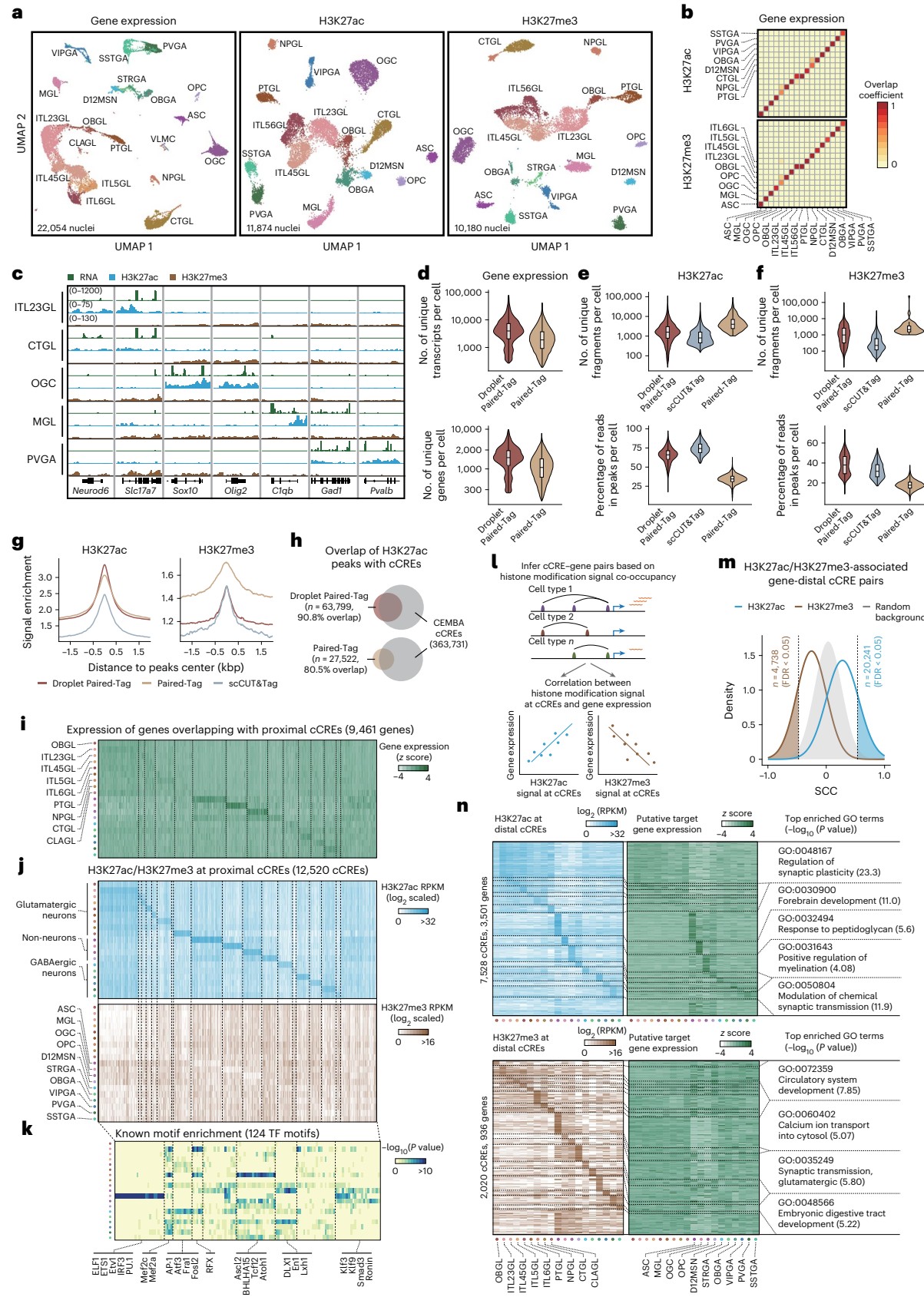

each cell cluster and retained those that appeared in two or more replicates. The resulting union set of 63,799 peaks was two times more than that detected with the previous combinatorial indexing-based Paired-Tag method (27,522) from a similar number of nuclei (11,874 versus 11,749; Extended Data Fig. 6d). A higher fraction of H3K27ac peaks detected in Droplet Paired-Tag overlapped with the open chromatin regions from the same brain cell types than the previous Paired-Tag dataset (90.8% versus 80.5%; Fig. 2h). Taken together, these results suggest that Droplet Paired-Tag can generate high-quality transcriptomic and epigenomic profiles at single-cell resolution from complex tissues.

To further demonstrate the utility of Droplet Paired-Tag in characterizing cCRE activity states, we examined variations of chromatin states (H3K27ac or H3K27me3) at identified cCREs in different brain cell types[23]. We classified cCREs as distal or proximal based on their distance to promoter regions (Methods) and performed non-negative matrix factorization to group all the cCREs with sufficient levels of H3K27ac (reads per kilobase (kb) per million (RPKM) > 1) and H3K27me3 (RPKM > 1) signals in at least one brain cell type into 20 cCRE modules, each showing a distinct pattern of cell-type specificity (Fig. 2i,j and Extended Data Fig. 8a). For proximal cCREs, H3K27ac signal in the promoter region showed a strong positive correlation with transcription, while H3K27me3 signal showed an overall inverse correlation (Fig. 2i). Transcription factor binding motif analysis of each CRE module revealed known transcription factors involved. For example, a cCRE module corresponding to the ITL23GL cluster (excitatory neurons from cortex layers 2 and 3) was enriched for NEUROD1 and MEF2C motifs, and the OGC (mature oligodendrocytes) cCRE module was enriched for motifs of oligodendrocyte-critical transcriptional factors, such as SOX10 (Fig. 2k, Extended Data Fig. 8b and Supplementary Table 5).

Droplet Paired-Tag data enable the prediction of putative target genes of distal cCREs due to the joint profiling of gene expression levels and chromatin states from the same cells. We calculated the pairwise Spearman's correlation coefficients (SCCs) between histone modification signals at distal cCREs and promoter regions of potential target genes within a 500-kb window (Fig. 2l). This analysis identified 20,241 significant cCRE–gene pairs with a positive correlation between H3K27ac signals and gene expression and 4,738 pairs with a negative correlation between H3K27me3 and gene expression (false discovery rate (FDR) of <0.05; Fig. 2m and Supplementary Table 6). Interestingly, Droplet Paired-Tag data captured stronger linkages between cCREs and genes over background than combinatorial indexing Paired-Tag (Extended Data Fig. 8c). Gene ontology (GO) analysis showed that H3K27ac-associated genes in the OGC population were enriched for terms related to the myelination process, consistent with the likely enhancer function of the distal cCREs. However, H3K27me3-associated genes in the same cell type were enriched for terms related to neuronal function (Fig. 2n and Supplementary Table 7). Through integrative analysis with a recently published mouse brain single-cell chromosome contacts dataset[24], we found that cell-type-specific cCRE–gene pairs with long-range chromatin contacts were overall more positively (for H3K27ac) correlated than cCRE–gene pairs with no detectable chromatin contacts (Extended Data Fig. 8d,e).

## Discussion

In summary, Droplet Paired-Tag is a fast and robust method for joint profiling of histone modifications and gene expression in single cells. We demonstrated the utility and superior performance of this method for analyzing cell-type-specific gene regulatory programs in complex tissues. By using a widely available microfluidic device (that is, 10x Genomics Chromium), this shortened, more easily adaptable procedure will likely facilitate the quick adaptation of this method in the field of epigenetics. Droplet Paired-Tag adds a new tool kit for investigation of the gene regulatory mechanisms in disease and life span.

## Online content

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

## Methods

### Cell culture

mESCs used in this study have been described in a previous study[20]. Specifically, a hybrid mouse embryonic cell line CAST x s129 was engineered to have both alleles of the *SOX2* gene tagged (CAST allele with eGFP; s129 allele with mCherry), and 4 copies of the CTCF binding sites inserted between *SOX2* and downstream super-enhancer on the CAST allele. mESCs were maintained in feeder-free and serum-free 2i medium at 37 °C with 5% $CO_2$. To isolate nuclei, mESCs were dissociated with Accutase (Innovative Cell Technologies, AT104), collected by centrifugation, washed twice with PBS (Gibco, 10010023) and resuspended in cold nuclei permeabilization buffer (10 mM Tris-HCl (pH 7.4; Sigma, T4661), 10 mM NaCl (Sigma, S7653), 3 mM $MgCl_2$ (Sigma, 63069), 1× protease inhibitor (Roche, 05056489001), 0.5 U μl$^{-1}$ RNaseOUT (Invitrogen, 10777-019), 0.5 U μl$^{-1}$ SUPERaseIn inhibitor (Invitrogen, AM2694), 0.1% IGEPAL CA630 (Sigma, I8896) and 0.02% digitonin (Sigma, D141)) for 1 min. Nuclei were counted using a Bio-Rad TC20 cell counter with 0.4% Trypan Blue (Gibco, 15250061) staining. For each Droplet Paired-Tag experiment, 0.5 million nuclei were used.

### Mouse brain dissection and nuclei extraction

All animal work described in this manuscript has been approved and conducted under the oversight of the University of California, San Diego, Institutional Animal Care and Use Committee. Male C57BL/6J mice were purchased from the Jackson Laboratory (000664) at 12 weeks of age and were housed in the barrier facility at University of California, San Diego, under a 12-h light/12-h dark cycle in a temperature-controlled room with ad libitum access to water and food until euthanasia and tissue collection at 16 weeks of age. The temperature in the animal facility was maintained within the range of 20 to 22.2 °C, while the humidity levels varied between 35 and 60%. The frontal cortex was dissected from 16-week-old male mice, snap-frozen in liquid nitrogen and stored at −80 °C before proceeding to nuclei extraction.

Single-cell suspensions were prepared from frozen tissues by Dounce homogenization in douncing buffer (0.25 M sucrose (Sigma, S7903), 25 mM KCl (Sigma, P9333), 5 mM $MgCl_2$, 10 mM Tris-HCl (pH 7.4), 1 mM DTT (Sigma, D9779), 1× protease inhibitor, 0.5 U μl$^{-1}$ RNaseOUT and 0.5 U μl$^{-1}$ SUPERaseIn inhibitor). The cell suspension was then filtered through a 30-μm Cell-Tric filter (Sysmex) for debris removal and centrifuged for 10 min at 300*g* at 4 °C. Cell pellets was washed once with douncing buffer, centrifuged again and resuspended in cold nuclei permeabilization buffer for 10 min. Permeabilized nuclei were pelleted by centrifugation for 10 min at 1,000*g* and 4 °C and washed with sort buffer (1× PBS (Gibco, 10010023), 1× protease inhibitor (Roche, 05056489001), 0.5 U μl$^{-1}$ RNaseOUT (Invitrogen, 10777-019), 0.5 U μl$^{-1}$ SUPERaseIn inhibitor (Invitrogen, AM2694), 1 mM EDTA (Invitrogen, 15575020) and 1% bovine serum albumin (BSA; Sigma, A1595)) once. After resuspension in sort buffer, nuclei were stained with 2 μM 7-AAD (Invitrogen, A1310) for 10 min on ice and were sorted by fluorescence-activated nuclei sorting with an SH800 cell sorter (Sony) for the isolation of single nuclei (Extended Data Fig. 9). Nuclei were collected in collection buffer (1× PBS (Gibco, 10010023), 5× protease inhibitor (Roche, 05056489001), 2.5 U μl$^{-1}$ RNaseOUT (Invitrogen, 10777-019), 2.5 U μl$^{-1}$ SUPERaseIn inhibitor (Invitrogen, AM2694), 1 mM EDTA (Invitrogen, 15575020) and 5% BSA (Sigma, A1595)) at 5 °C and immediately centrifuged for 10 min at 750*g* and 4 °C. Nuclei were washed twice with sort buffer and counted on an RWD C100-Pro fluorescence cell counter with DAPI staining to better estimate the number of nuclei. For each histone modification, around 0.5 million sorted nuclei were aliquoted and used for Paired-Tag experiments.

### Assembly of the active transposon complex

Sequences for all custom oligonucleotides used in this study are provided in Supplementary Table 1. To assemble pA–Tn5 complexes suitable for the 10x Single Cell Multiome ATAC + Gene Expression platform, we mixed transposome oligonucleotides (100 μM) in two separate PCR tubes (USA Scientific, 1402-2300) with 25 μl of AdapterA + 25 μl of bridge-pMENTs or 25 μl of AdapterB + 25 μl of pMENTs. Oligonucleotides were annealed in a thermal cycler with the following program: 95 °C for 5 min and slowly cool to 12 °C at a speed of 0.1 °C s$^{-1}$. Each 1 μl of the annealed transposome DNA was mixed separately with 6 μl of unloaded pA–Tn5 (0.5 mg ml$^{-1}$, MacroLab) and quickly spun down. The mixtures were incubated at room temperature for 30 min then at 4 °C for an additional 10 min, and equal volumes of assembled pA–Tn5–AdapterA and pA–Tn5–AdapterB were mixed to form functional pA–Tn5 complexes. Assembled transposon complexes can be stored at −20 °C for up to 6 months, and transposon activity was validated with a bulk CUT&Tag assay before use.

### Antibodies

Antibodies used in this study include H3K27ac (Abcam, ab177178, recombinant; Abcam, ab4729, polyclonal) and H3K27me3 (Abcam, ab192985, recombinant). We found that antibody specificity is critical for high-quality signals of single-cell histone data. For H3K27ac, although the recombinant antibody yielded a higher fragment number per cell than the polyclonal antibody, its enrichment at TSSs or ChIP–seq peaks was lower (Extended Data Fig. 6e). Therefore, except for replicate 1 of the mouse frontal cortex dataset, all other experiments targeting H3K27ac were performed with the polyclonal antibody. One microgram of antibody was used per Droplet Paired-Tag reaction.

### Experimental protocol for Droplet Paired-Tag

A brief description of the Droplet Paired-Tag experimental procedure is provided below. A more detailed, step-by-step protocol is provided in Supplementary Data 1 and on the Protocol Exchange[25].

**Antibody-guided tagmentation.** pA–Tn5 and primary antibody were preconjugated during nuclei extraction. One microgram of primary antibody and 1 μl of assembled pA–Tn5 were premixed in 35 μl of MED1 buffer (20 mM HEPES (pH 7.5), 300 mM NaCl, 0.5 mM spermidine, 1× protease inhibitor cocktail, 0.5 U μl$^{-1}$ SUPERaseIn RNase inhibitor, 0.5 U μl$^{-1}$ RNaseOUT, 0.01% IGEPAL CA630, 0.01% digitonin, 2 mM EDTA and 1% BSA) and rotated at room temperature for 1 h, as a previous study showed that high-salt conditions are critical to reducing undesired open chromatin background[9]. Nuclei extracted with the above-described protocol were also resuspended in MED1 buffer, and 0.15 million–0.50 million nuclei were distributed into the premixed antibody and pA–Tn5 to a final volume of 75 μl. The mixtures were rotated at 4 °C overnight for epitope targeting along with pA–Tn5 tethering.

After overnight incubation, nuclei were isolated by centrifugation for 10 min at 300*g* and 4 °C and washed with MED2 buffer (20 mM HEPES (pH 7.5), 300 mM NaCl, 0.5 mM spermidine, 1× protease inhibitor cocktail, 0.5 U μl$^{-1}$ SUPERaseIn inhibitor, 0.5 U μl$^{-1}$ RNaseOUT, 0.01% IGEPAL CA630, 0.01% digitonin and 1% BSA) three times to remove excess antibody and pA–Tn5. Tagmentation was next performed in 50 μl of MED2 buffer supplemented with 10 mM $MgCl_2$ (Sigma, M1028) at 550 r.p.m. and 37 °C for 60 min in a ThermoMixer (Eppendorf). The tagmentation reaction was terminated by adding an equal volume of stop solution (10 mM Tris-HCl (pH 8.0), 20 mM EDTA, 2% BSA, 2× protease inhibitor cocktail, 1 U μl$^{-1}$ SUPERaseIn inhibitor and 1 U μl$^{-1}$ RNaseOUT). Nuclei were spun down for 10 min at 500*g* and 4 °C and washed once with 1× nuclei buffer (10x Genomics, PN-2000207; supplemented with 1 mM DTT, 0.5 U μl$^{-1}$ SUPERaseIn inhibitor and 0.5 U μl$^{-1}$ RNaseOUT). Finally, nuclei were resuspended in 10 μl of 1× nuclei buffer, and 10,000–16,000 nuclei were aliquoted into PCR tubes with a total volume of 8 μl. Normally, the nuclei recovery rate is around 30–60% depending on the starting input materials. To assess CUT&Tag performance, around 10,000–50,000 nuclei were used for bulk analysis.

Fragmented DNA in these nuclei was purified with MiniElute (Qiagen, 28004) columns and amplified for quality control assessment. Seven microliters of ATAC Buffer B (10x Genomics, PN-2000193) was added to 8 µl of nuclei mixture to reach a final reaction volume of 15 µl, the same as specified in the user manual of the Chromium Next GEM Single Cell Multiome kit (CG000338, RevF), except that we substituted 3 µl of ATAC Enzyme B (10x Genomics, PN-2000265/PN-2000272) with 1× nuclei buffer.

**Reverse transcription, cell barcoding and library preparation.** Barcoding reaction mixtures were prepared as described in the manual for the Chromium Next GEM Single Cell Multiome kit. Sixty microliters of prepared master mix was added to 15 µl of nuclei mixture before being loaded onto a Chromium Next GEM Chip J and proceeding to droplet generation with a Chromium X microfluidic system (10x Genomics). Reverse transcription and cell barcoding were performed inside 10x GEM. Final DNA and RNA library amplification was performed according to the Chromium Single Cell ATAC Library kit manual except that we used an increased number of amplification cycles (12–13 in total) for histone modality libraries.

### Preprocessing of Droplet Paired-Tag data

All sequencing was performed with an Illumina Nextseq2000 sequencer. Droplet Paired-Tag fastq files were demultiplexed using cellranger-arc (v2.0.0) with the command 'cellranger-arc mkfastq'; however, DNA and RNA data were preprocessed using cellranger-atac (v2.0.0) and cellranger (v6.1.2), respectively, and barcodes were manually paired with a custom script using the matching relationship provided in cellranger-arc. To select high-quality nuclei, we first aggregated histone modification data from the same sample and performed peak calling to find narrow peaks (for H3K27ac) or broad peaks (for H3K27me3) using MACS2 (v2.1.2)[26]. We then filtered the histone modality based on the per cell fragment number and FRiP. We next selected nuclei by pairing pass-filtered nuclei from both modalities (histone modification and transcriptome), as shown in Extended Data Fig. 3a. Before clustering, nuclei with a high fraction of mitochondrial and ribosomal RNA reads were filtered out. Nuclei with an extremely high number of reads were also filtered out because most of them were doublets. Potential doublets were identified and removed using Scrublet[27] for individual RNA datasets, and corresponding DNA barcodes were also removed.

For genome browser track generation, sample- or cell-type-specific bigwig files were generated from bam files with deepTools (v3.5.1)[28] and visualized in Integrative Genomics Viewer (v2.15.4)[29].

For FRiP calculation, duplicate reads were removed using Samtools (v1.14)[30] and Picard MarkDuplicates (v2.25.0)[31], taking barcode information into account. Peak calling was then performed using MACS2 with default parameters, except that for H3K27me3, we called broad peaks with '–broad' due to its broad domain enrichment. Preprocessing pipelines and scripts are shared at https://github.com/Xieeeee/Droplet-Paired-Tag.

### Analysis of Droplet Paired-Tag data

**Signal enrichment calculation.** Density plots and heat maps of signal enrichment over ChIP–seq peaks or CEMBA cCREs were generated using deepTools. Peaks that overlapped with ENCODE blacklist (v2) or CUT&RUN blacklist regions were removed during the calculation of enrichment[32,33].

**Clustering and annotation.** Clustering of single-cell transcriptomic data was performed in R using Seurat (v4.1.0)[34] and Signac (v1.6.0)[35]. In short, gene counts were normalized, and the top 2,500 variable genes were selected for dimension reduction by principal-component analysis. For all datasets, the first 35 principal components were used to correct batch effects with Harmony[36], followed by UMAP visualization and Louvain clustering. Marker genes for each cluster were identified using Seurat, with the $\log_2$ (fold change) threshold set to 0.25. Annotation of cluster identities was done with marker genes characterized in previous studies. For epigenomic data, 10x fragment files were converted to cell-by-bin matrices using 5-kb non-overlapping genomic bins, and clustering was performed using Signac. Briefly, sequencing depth was normalized using the two-step term frequency-inverse document frequency. The top 85% of genomic bins were selected for linear dimension reduction by singular value decomposition, and batch effects were corrected with Harmony, again followed by UMAP visualization and Louvain clustering. Gene activity scores were computed by signal density in promoter and gene body regions.

To compare clustering results from different modalities, we first annotated the epigenome clusters by nominating the most abundant cell type identified with transcriptome clustering. Overlap coefficients ($O_i$) were calculated according to the proportion of cells sharing the same labels from both the transcriptome ($A$) and epigenome clusters ($B$) in the transcriptome clusters:

$$O_i = \max\left(\frac{A_x \cap B_i}{A_i}\right).$$

**Integration with public snRNA-seq datasets.** Integration of the Droplet Paired-Tag RNA dataset with the original Paired-Tag dataset and 10x snRNA-seq dataset was performed using Seurat. Briefly, gene counts for all datasets were normalized, and the top 2,000 shared variable genes across datasets were identified as integration features. Dimensional reduction (canonical correlation analysis) was performed to project all nuclei into the same embedding, and 'anchors' (pairs of cells from different datasets) were identified by mutual nearest neighbors searching. Low-confidence anchors were filtered out, and shared neighbor overlap between anchor and query cells in an overall neighbor graph was computed. Louvain clustering on the overall neighbor graph was used for coembedded cluster identification. To compare clustering results from different datasets, overlap coefficients ($O_{i,j}$) were calculated based on the number of cells sharing the same labels from both the query ($A$) and reference clusters ($B$) in the coembedding clusters ($C$; $i$ indicates query cell type, $j$ indicates reference cell type, and $k$ indicates coembedding cluster):

$$O_{i,j} = \min\left(\left[\frac{A_i \cap C_k}{A_i}\right], \max\left[\frac{B_j \cap C_k}{B_j}\right]\right).$$

**Identification of peak set.** To identify peaks using H3K27ac data, we adopted a previously described method to perform peak calling and merge peaks across replicates[23]. Properly paired reads from all pass-filtered nuclei in the same replicate were merged to generate a pseudo-bulk dataset for all biological replicates. After shift correction for the pA–Tn5 cleavage site, peak calling was performed using MACS2 with the following parameters: '-q 0.01 –nomodel –shift −75 –extsize 150 –keep-dup all -B –SPMR'. Because we used two different antibodies to reduce variation caused by antibody affinity and specificity, we retained peaks identified in at least two replicates as conserved peaks during merging. Finally, we extended peak summits to a fixed width of 500 base pairs for merging and downstream analysis.

**Identification of cCRE modules.** To ensure a fair comparison between different techniques, we filtered the CEMBA cCREs list for elements with an arbitrary cutoff (histone modification signal RPKM of >1) in at least one transcriptome-based cluster from both Droplet Paired-Tag and Paired-Tag datasets and retained cCREs with H3K27ac (289,437, 87.9% of cCREs) or H3K27me3 (127,005, 34.9% of cCREs) signal RPKM values of >1 in at least one cluster for downstream analysis. For visualization, shared cCREs (108,319) from H3K27ac and H3K27me3 groups were selected for plotting. cCREs were classified as distal or proximal

based on their distance to ±2 kb of the TSS in GENCODE mm10 (vm25). Proximal or distal cCREs were grouped into 20 different modules by non-negative matrix factorization[37] based on their histone modification signal intensity. Downstream motif enrichment and GO analysis are based on this classification of cCRE modules. For visualization, 95,799 distal cCREs and 12,520 proximal cCREs that passed both H3K27ac and H3K27me3 signal cutoffs were plotted. Cell types with <100 nuclei were excluded for further analysis (vascular and leptomeningeal cells).

**Linking cCREs with putative target genes.** We used a previously described method to link cCREs with their putative regulatory genes for both histone modifications[38]. First, cCREs with co-occupancy of H3K27ac or H3K27me3 within a genomic distance of 500 kb were identified using Cicero[39]. cCREs with co-occupancy (Cicero score) of >0.1 were retained for further analysis. Next, cCREs were classified as distal or proximal based on their distance to ±2 kb of the TSS in GENCODE mm10 (vm25). In our analysis, only distal-to-proximal pairs were selected for comparison. We calculated SCCs between gene expression and histone modification signal over cCRE across clusters to examine the relationship between coaccessibility pairs. To estimate random background levels, we shuffled the cell identities for each read and calculated the corresponding SCCs. Finally, we fit a normal distribution model and set a cutoff SCC score with an FDR of <0.05 as an empirically defined significance threshold to select significant positively (H3K27ac) or negatively (H3K27me3) correlated cCRE–gene pairs. To compare to the original Paired-Tag dataset on the strength of putative cCRE–gene pairs, we used a Kolmogorov–Smirnov test to calculate the difference between putative cCRE–gene linkages over random background. For comparison to the previously published Paired-Tag dataset, the greatest distance ($D$) between real and background distributions was calculated for each dataset, respectively.

**Motif enrichment and GO analysis.** Motif enrichment for each cCRE module was performed using HOMER (v4.11) and the 'findMotifsGenome.pl' function[40]. The displayed motif heat maps in Fig. 2k and Extended Data Fig. 8b were from the results of known motif discovery. GO analysis for each enriched gene set was performed using PANTHER with default parameters, and biological process terms were used for annotation[41]. To exclude ambiguous terms, we only selected the top enriched terms ranked by fold enrichment × $-\log_{10}$ (adjusted $P$ value).

**Integrative analysis of Droplet Paired-Tag and single-nucleus methyl-3C sequencing (snm3C-seq) data.** The mouse brain snm3C-seq dataset was downloaded from Gene Expression Omnibus (GEO) with accession number GSE156683 (ref. 24). Contact pairs from individual cells were merged and visualized using pairtools (v1.0.2) at a 5-kb resolution[42]. To summarize putative cCRE–gene pairs at loop anchors, we first performed loop calling using HiCCUPS (Juicer tools v1.22.01) at resolutions of 5, 10 and 25 kb (ref. 43). Merged loop sets were then intersected with cCRE–gene pairs using bedtools pairtobed (v2.27.1)[44].

**Reporting summary**
Further information on research design is available in the Nature Portfolio Reporting Summary linked to this article.

## Data availability
Raw data obtained in this study have been deposited at NCBI GEO (http://www.ncbi.nlm.nih.gov/geo/) with accession number GSE224560. The processed data can also be accessed as supplementary files in GEO. Datasets for mESC H3K27ac ChIP–seq were downloaded from the 4DN data portal with the accession number 4DNESTVGLCD9. Other external datasets were downloaded from NCBI GEO with the following accession numbers: mESC H3K27me3 ChIP–seq data (GSE156589), Paired-Tag data (GSE152020), CoTECH data (GSE158435), scCUT&Tag data from

peripheral blood mononuclear cells (GSE157910), scCUT&Tag data from brain (GSE163532), scCUT&Tag-pro data (GSE195725), snm3C-seq data from brain (GSE156683) and ChIP–seq data from mouse cortex excitatory neurons (GSE141587). The BICCN single-nucleus ATAC-seq datasets and BICCN 10x snRNA-seq MOp data were downloaded via the NeMO archive (RRID SCR_016152; https://assets.nemoarchive.org/dat-ch1nqb7). The 10x peripheral blood mononuclear cell scRNA-seq and E18 embryonic mouse brain Multiome datasets were downloaded from the 10x Genomics website (https://www.10xgenomics.com/resources/datasets). Source data are provided with this paper.

## Code availability
Scripts and code are available at https://github.com/Xieeeee/Droplet-Paired-Tag.

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

## Acknowledgements

We thank K. Zhang and E. Armand for discussions and suggestions for data analysis. We thank QB3 MacroLab for pA–Tn5 fusion protein production. This study was funded by the National Institutes of Health (grant numbers R41MH128993 and RF1MH128838) and the Ludwig Institute for Cancer Research (to B.R.), and the National Institutes of Health grant numbers 4R00HG011483 and 5RM1HG011014 (to C.Z.). Z.W. is a DDBrown Awardee of the Life Science Research Foundation.

## Author contributions

B.R., Y.X. and C.Z. conceived and designed the study and wrote the paper. Y.X. performed the Droplet Paired-Tag experiments. Y.X., C.Z. and Y.E.L. performed the data analysis. Z.W. collected and dissected the mouse brain tissues. M.T. provided the mESCs. Z.W. performed the combinatorial indexing-based Paired-Tag on mESCs. Y.X. and L.C. designed and tested different bridge oligonucleotides for the 10x Genomics Multiome platform. B.R. supervised the research. All authors discussed the results and edited the paper.

## Competing interests

B.R. is a cofounder and consultant for Arima Genomics, Inc., and a cofounder of Epigenome Technologies, Inc. B.R. and C.Z. are listed as inventors on a patent application titled 'Parallel analysis of individual cells for RNA expression and DNA from targeted tagmentation by sequencing'. All other authors declare no competing interests.

## Additional information

**Extended data** is available for this paper at https://doi.org/10.1038/s41594-023-01060-1.

**Correspondence and requests for materials** should be addressed to Bing Ren.

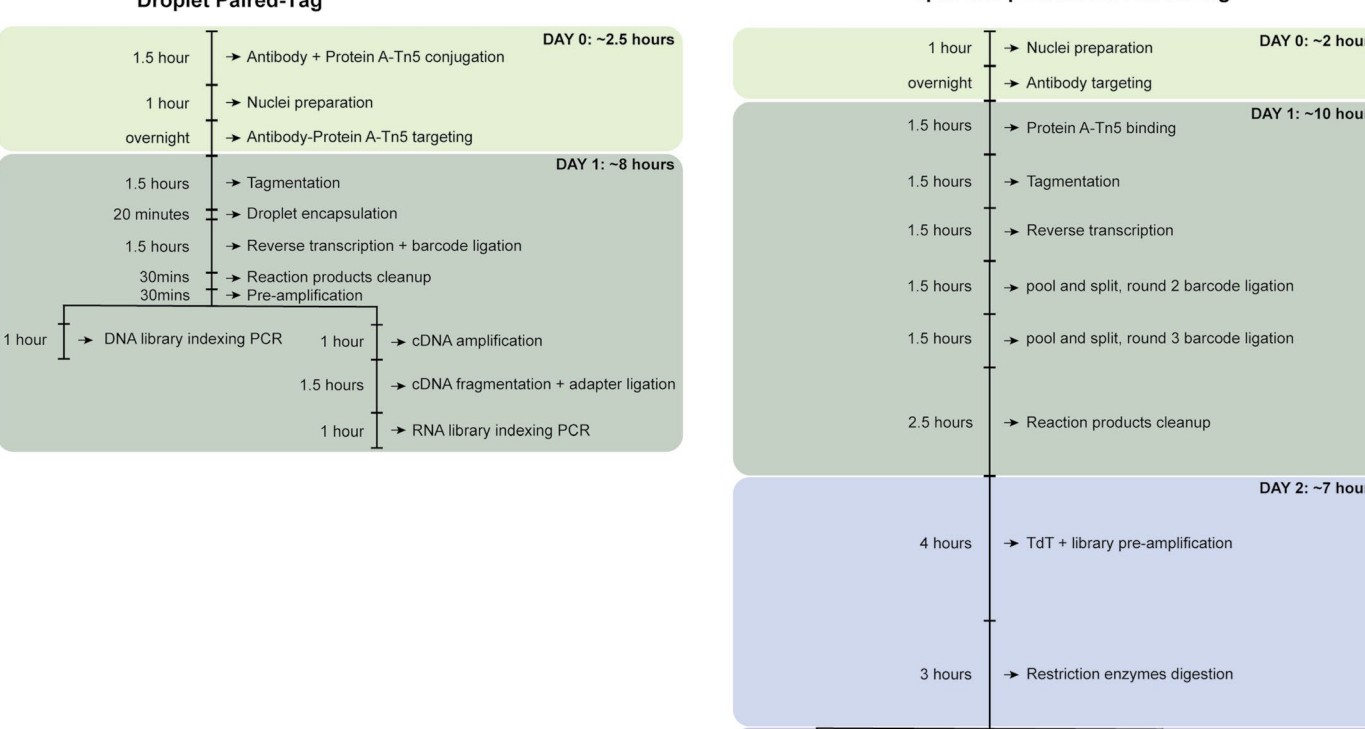

**Extended Data Fig. 1 | Comparison of the workflows of Droplet Paired-Tag and combinatorial-indexing-based Paired-Tag.** Detailed workflow describing the end-to-end procedure of Droplet Paired-Tag (left) is compared to the conventional combinatorial-indexing-based version of Paired-Tag (right).

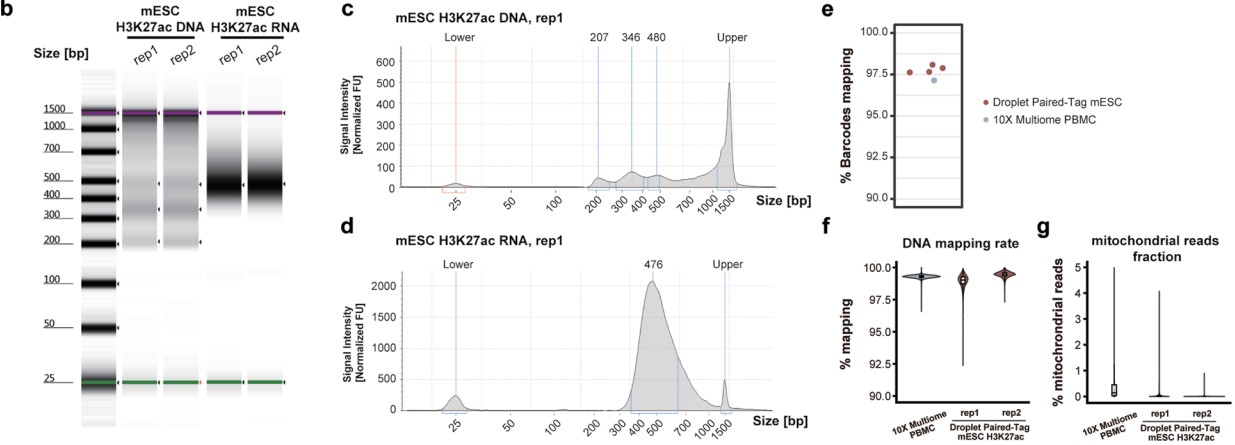

**Extended Data Fig. 2 | See next page for caption.**

**Extended Data Fig. 2 | Experimental design and quality control metrics of Droplet Paired-Tag. a**, Schematics for Droplet Paired-Tag barcoding and library preparation process. Modification on oligos is not shown for simplicity. **b**–**d**, Example capillary electrophoretic gel images from a TapeStation showing final libraries size distribution for Droplet Paired-Tag DNA (**c**) and RNA (**d**). All replicates are shown in **b**. **e**, Barcode mapping rate for Droplet Paired-Tag

and standard 10X Multiome experiment. **f**,**g**, Sequence mapping rate (**f**) and fraction of mitochondrial reads (**g**) for Droplet Paired-Tag DNA libraries. For all boxplots, hinges were drawn from the 25th to 75th percentiles, with the middle line denoting the median, whiskers with maximum 2 interquartile range (IQR). n = 4,302 (Droplet Paired-Tag, H3K27ac, replicate#1), 9,346 (Droplet Paired-Tag, H3K27ac, replicate#2), 2,711 (10X Multiome, ATAC).

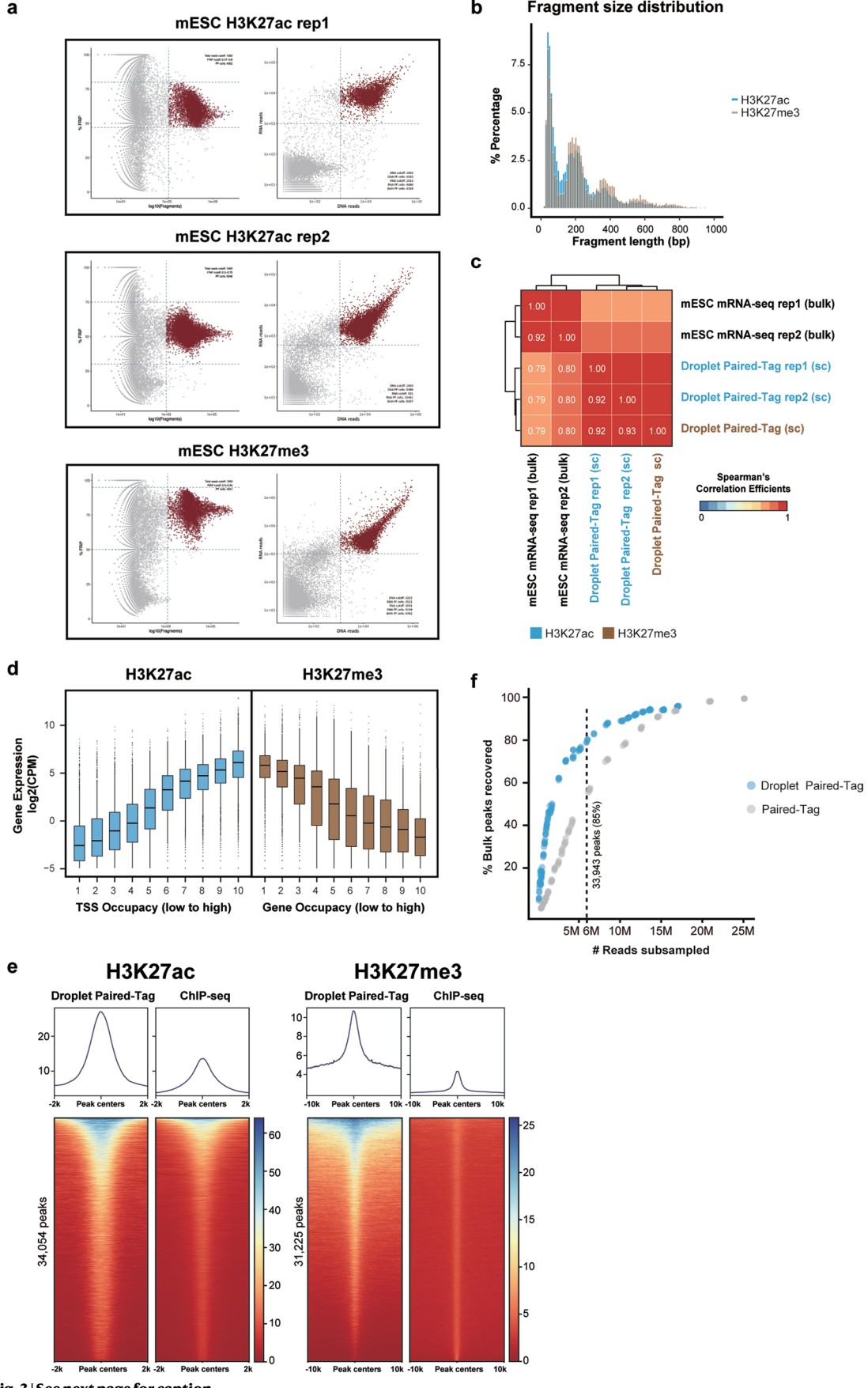

**Extended Data Fig. 3 | See next page for caption.**

**Extended Data Fig. 3 | Quality control metrics of mESC Droplet Paired-Tag data. a**, Strategies of valid nuclei selection. DNA barcodes were filtered based on total reads per nuclei and fraction of reads in peak regions. Valid nuclei were further selected based on the pairing of valid DNA and RNA nuclei in the scatterplot of total reads per nuclei (right). Cutoffs were set by manual inspection and depicted as dash lines and are also annotated inside the scatterplot. **b**, Distribution for fragment lengths of the sequenced fragments from the H3K27ac and H3K27me3 Droplet Paired-Tag experiments. **c**, Heatmap showing pairwise Spearman's correlation coefficients of expression profiles from single-cell Droplet Paired-Tag experiment and with the bulk mRNA-seq datasets. **d**, Boxplots showing the expression level of genes grouped by histone marks

occupancy at their TSS regions (H3K27ac) or gene bodies (H3K27me3). For all boxplots, hinges were drawn from the 25th to 75th percentiles, with the middle line denoting the median, whiskers with maximum 1 interquartile range (IQR), outlier indicated with dots. For H3K27ac group 1–10, gene number n = 22,456 (2,246 each quantile group except group 10); for H3K27me3 group 1–10, n = 22,969 (2,297 each quantile group except group 10). **e**, Enrichment of histone modification signals over ChIP-seq peaks, compared between Droplet Paired-Tag and ChIP-seq data. **f**, Scatter plot showing the relationships between total number of H3K27ac peaks called and the number of reads from Droplet Paired-Tag or Paired-Tag dataset in a down-sampling test, similar to Fig. 1g. Dashed line indicates 85% peaks recovered for Droplet Paired-Tag dataset.

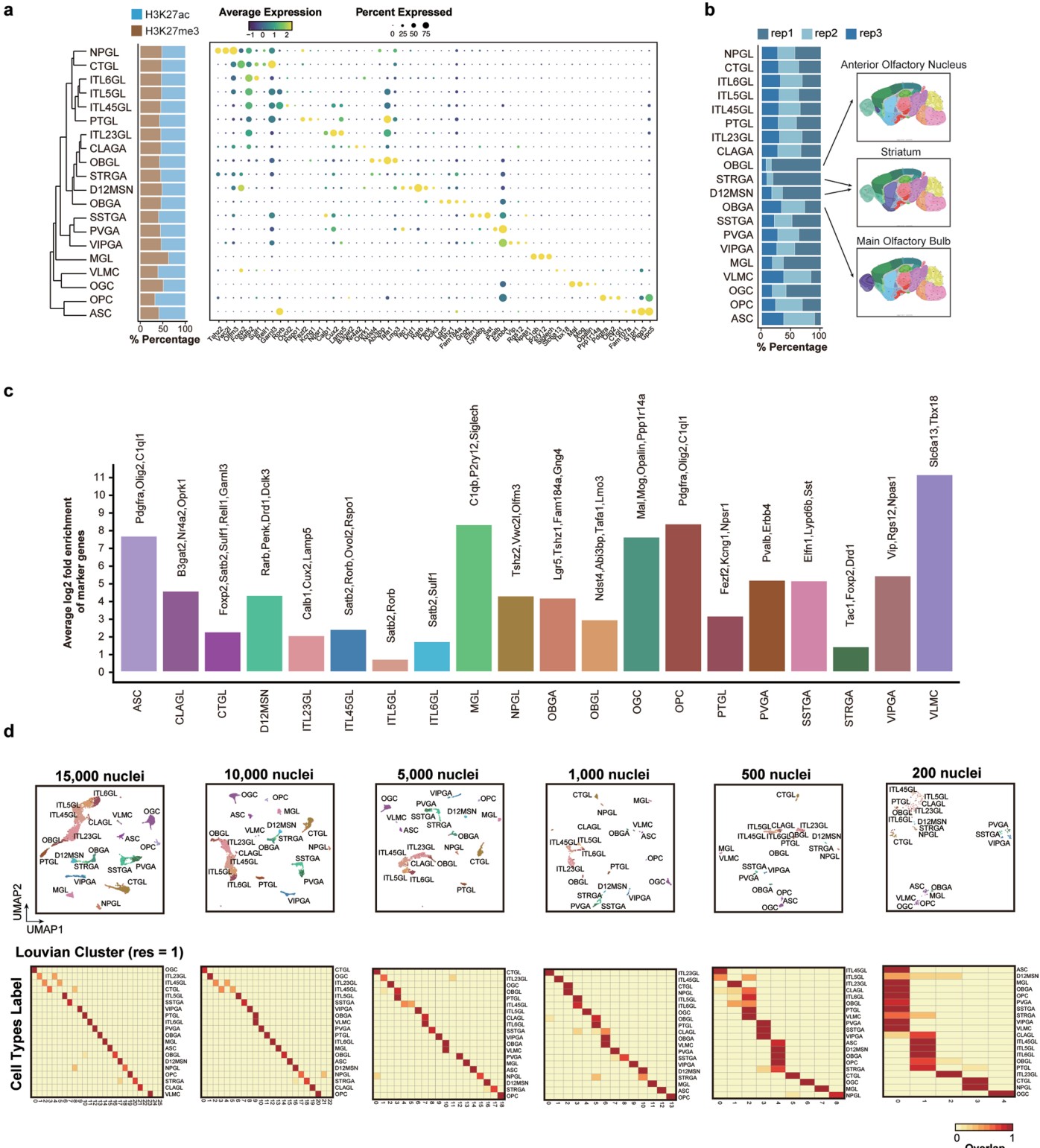

**Extended Data Fig. 4 | Annotation of cell clusters in the mouse frontal cortex Droplet Paired-Tag based on transcriptomic profiles. a**, Expression of marker genes in 20 mouse brain cell types, and the fraction of nuclei by set of experiments. **b**, Fraction of nuclei in each cell type by replicates. Cell types with biased distribution in any of the replicates were from anatomical regions (right) proximal to frontal cortex. **c**, Averaged Log$_2$ fold enrichment ratio of the normalized selected marker genes expression (RPKM) in cluster of interest versus the rest of the dataset. **d**, UMAP embeddings and overlap scores based on Droplet Paired-Tag transcriptome profiles down-sampled to different nuclei numbers.

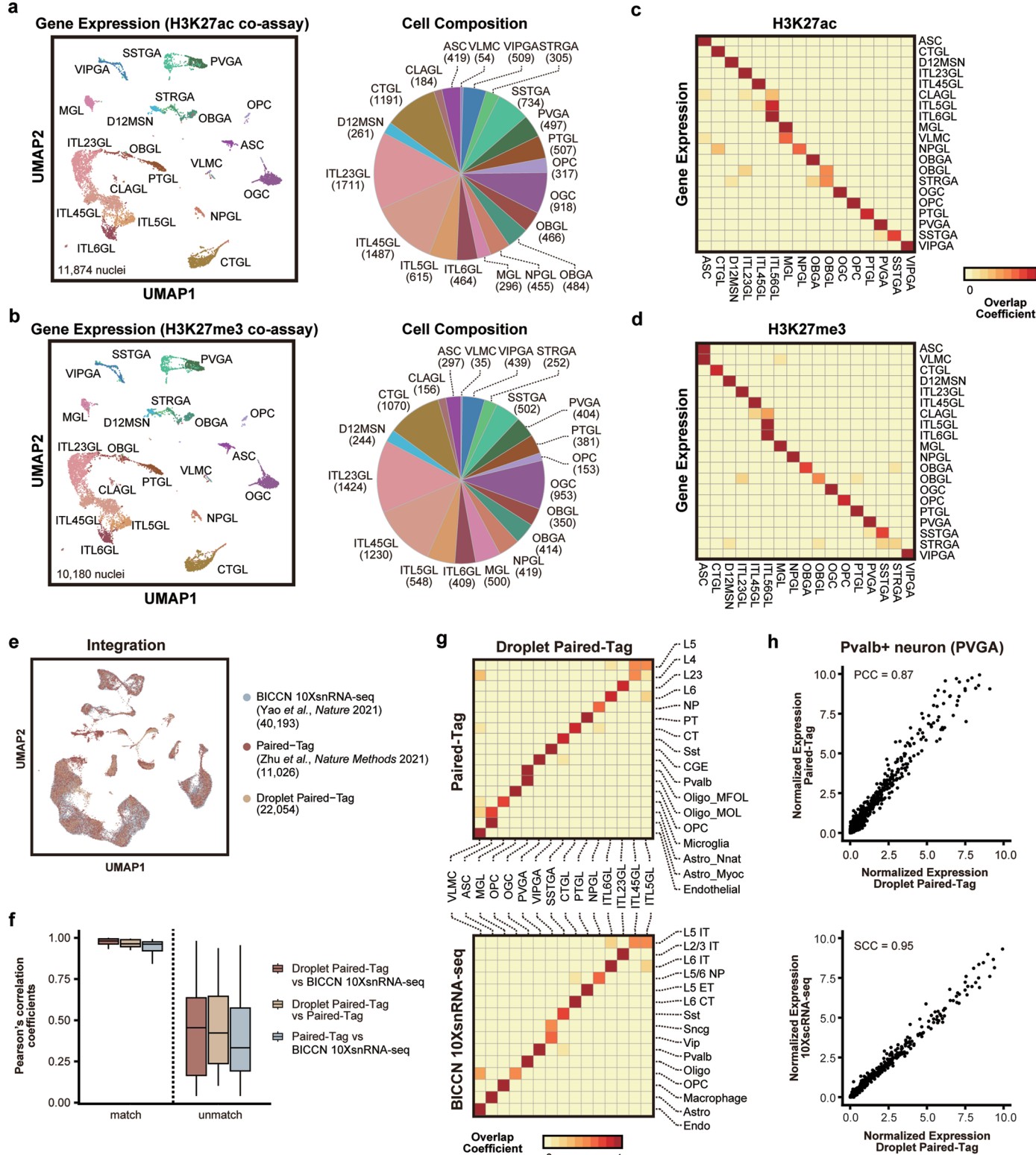

**Extended Data Fig. 5 | Integrative analysis of Droplet Paired-Tag transcriptomic profiles with public datasets. a**, **b**, UMAP embedding and cell type compositions of Droplet Paired-Tag transcriptome profiles based on histone modification co-profiled. **c**,**d**, Overlap of all annotations between transcriptomic and epigenomic clustering. **e**, UMAP co-embedding of single nuclei transcriptomic profile from Droplet Paired-Tag, Paired-Tag and reference BICCN10X snRNA-seq datasets on mouse motor cortex regions. **f**, Boxplot showing Pearson correlation coefficients of variable genes for matched and unmatched cell types between Droplet Paired-Tag, Paired-Tag and reference 10X snRNA-seq datasets. For all boxplots, hinges were drawn from the 25th to 75th percentiles, with the middle line denoting the median, whiskers with maximum 1.5 interquartile range (IQR). Number of matched cell types: n = 16 (Droplet Paired-Tag vs snRNA-seq), 18 (Droplet Paired-Tag vs Paired-Tag), 16 (Paired-Tag vs snRNA-seq). Number of unmatched cell types: n = 304 (Droplet Paired-Tag vs snRNA-seq), 322 (Droplet Paired-Tag vs Paired-Tag), 256 (Paired-Tag vs snRNA-seq) **g**, Overlap of shared annotations between Droplet Paired-Tag and the previously published Paired-Tag transcriptomic clustering results. Cell types not from frontal cortex are excluded in comparison. **h**, Scatterplots showing expression levels of variable genes in *Pvalb+* neurons (PVGA) in Droplet Paired-Tag, Paired-Tag and reference 10X snRNA-seq datasets.

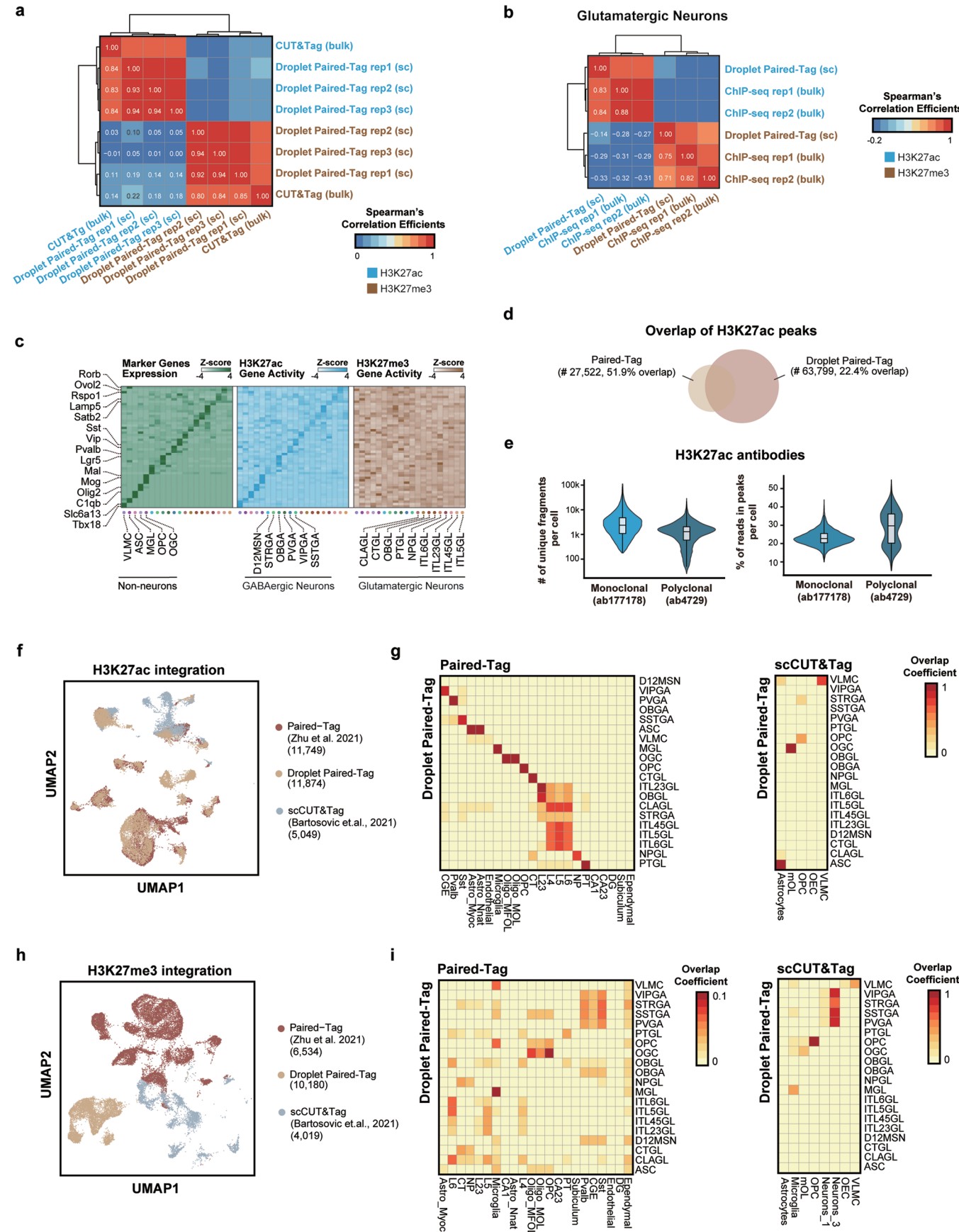

**Extended Data Fig. 6 | See next page for caption.**

**Extended Data Fig. 6 | Quality control of mouse frontal cortex Droplet Paired-Tag histone modifications profiles. a**, Genome-wise Spearman's correlation coefficients between mouse frontal cortex histone modification datasets from single-cell Droplet Paired-Tag and bulk CUT&Tag. **b**, Genome-wise Spearman's correlation coefficients between mouse frontal cortex glutamatergic neurons histone modification datasets from single-cell Droplet Paired-Tag and bulk ChIP-seq. **c**, Heatmap showing marker genes expression level and promoters / gene bodies histone modifications signal in each cell type. Examples of well-known marker genes are shown. **d**, Overlap of the number of H3K27ac peaks called from Droplet Paired-Tag and Paired-Tag datasets. Both datasets are down-sampled to similar number of cells. **e**, Comparison of the number of unique fragments and FRiP in single cell between monoclonal and polyclonal H3K27ac antibodies used. For all boxplots, hinges were drawn from the 25th to 75th percentiles, with the middle line denoting the median, whiskers with maximum 2 interquartile range (IQR). n = 8,807 (monoclonal antibody), 10,069 (polyclonal antibody). **f**, UMAP co-embedding of single nuclei H3K27ac profile from Droplet Paired-Tag, Paired-Tag and scCUT&Tag. **g**, Overlap of shared annotations between Droplet Paired-Tag and public single-cell H3K27ac datasets. **h**, UMAP co-embedding of single nuclei H3K27me3 profile from Droplet Paired-Tag, Paired-Tag and scCUT&Tag. **i**, Overlap of shared annotations between Droplet Paired-Tag and public single-cell H3K27ac datasets.

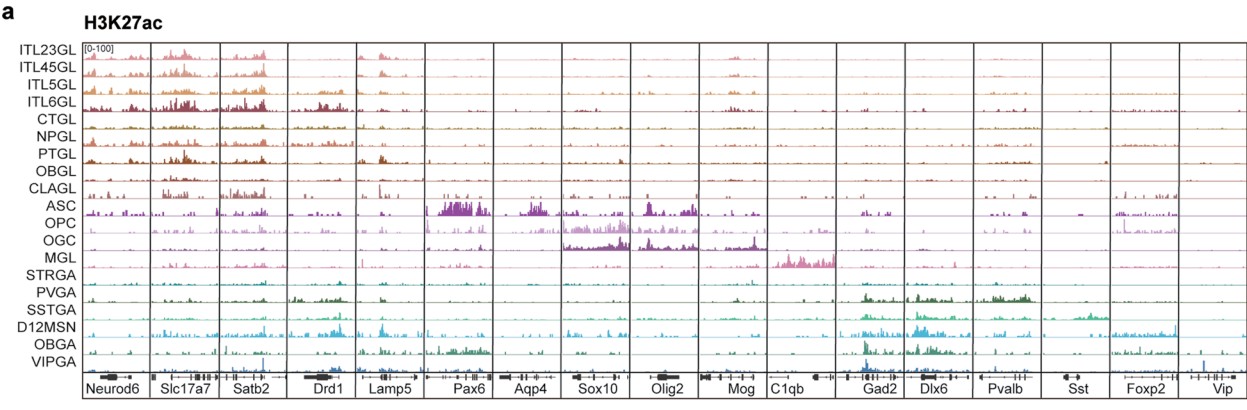

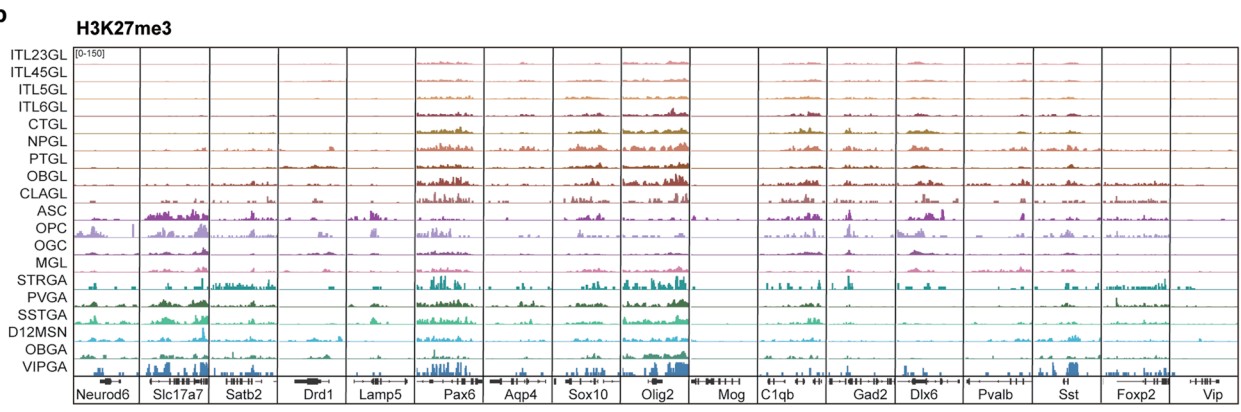

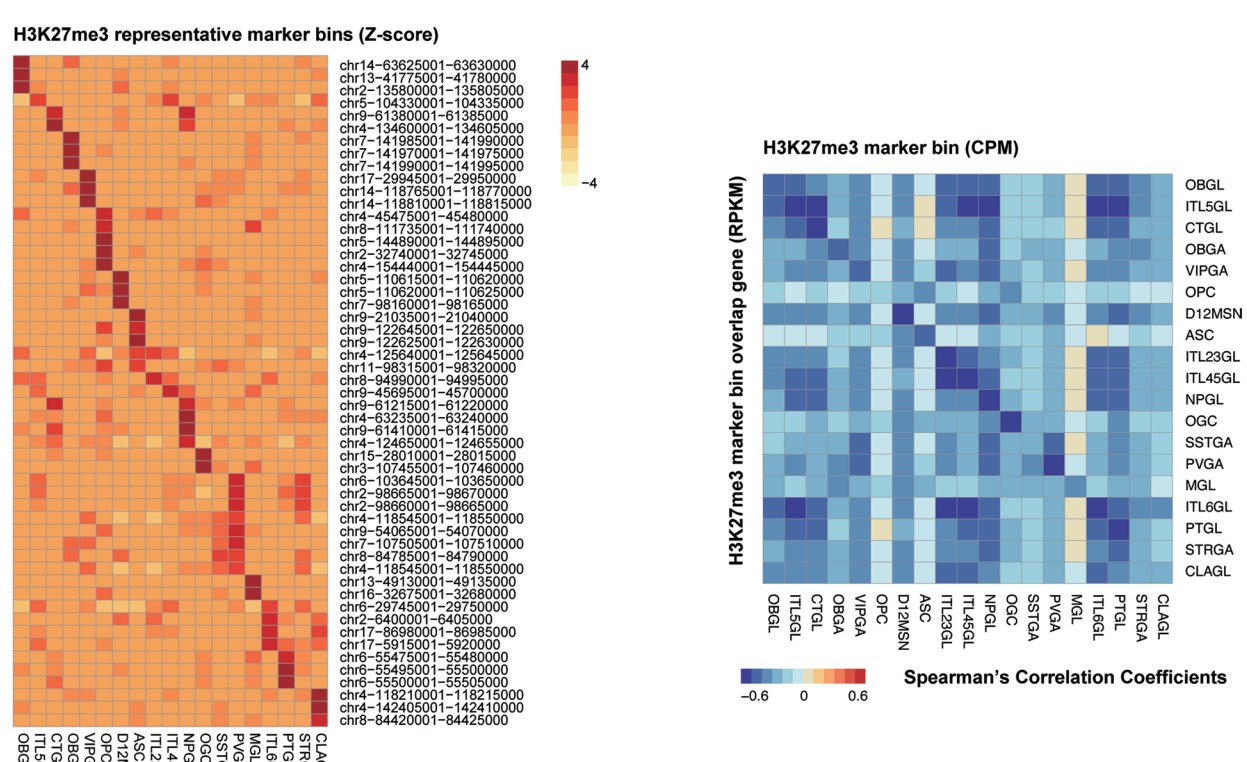

**Extended Data Fig. 7 | Landscape of histone modifications across cell types in mouse frontal cortex. a**, **b**, Representative Genome browser view showing H3K27ac (**a**) or H3K27me3 (**b**) signal over marker genes in each mouse brain cell type. **c**, Top 10 H3K27me3 marker bins from each cell type with >100 nuclei profiled. **d**, Spearman's correlation coefficients between cell-type specific marker bins H3K27me3 signal (CPM) and bin-overlapped genes expression level (RPKM) from different cell types.

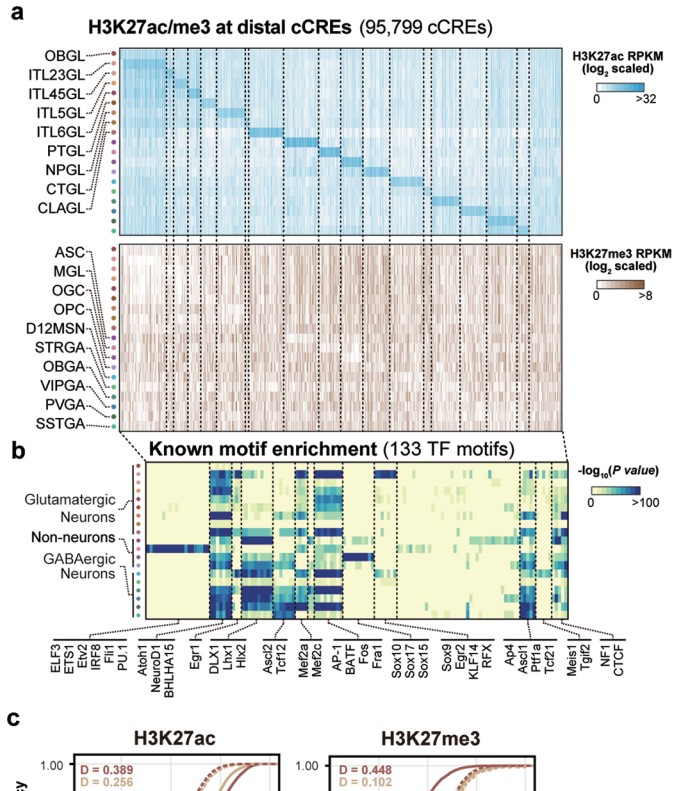

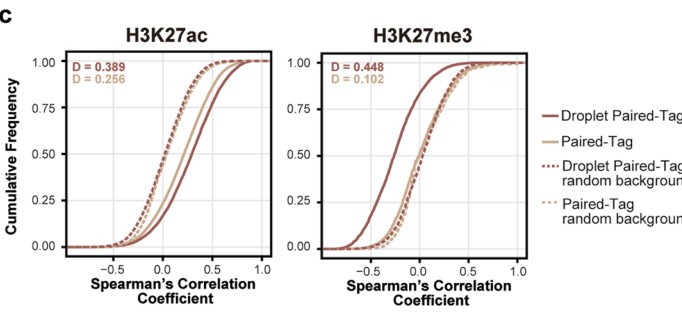

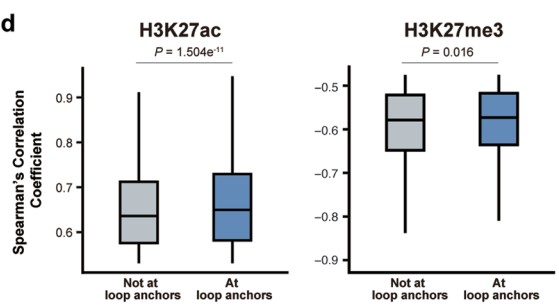

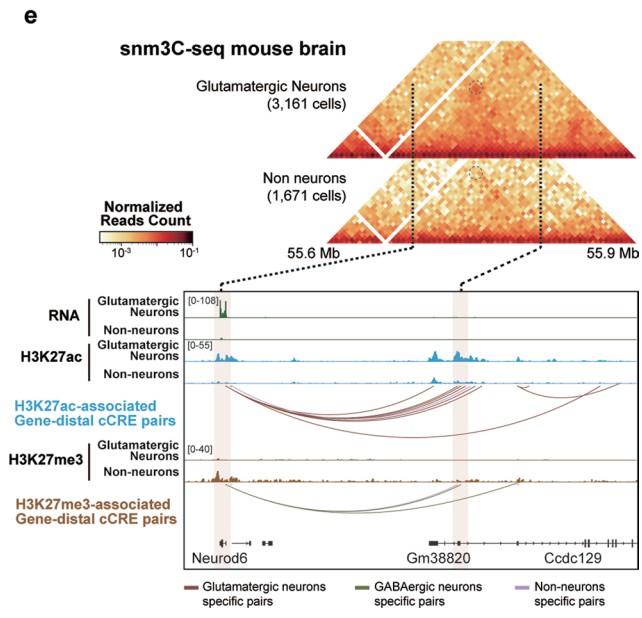

**Extended Data Fig. 8 | Integrative analysis of histone modifications at candidates *cis*-regulatory elements across cell types in mouse frontal cortex.** **a**, Heatmap showing H3K27ac and H3K27me3 signals at the distal cCREs across different cell types. **b**, Heatmap of known motifs enrichment for each cCRE module of distal cCREs. Examples of known motifs are shown along with the heatmap. *P* value: one-sided binomial test. False Discovery Rate is then calculated to select enriched motifs. **c**, Cumulative distribution function plot of Spearman's correlation coefficients between distal cCREs histone modification signal and putative target genes expression level. Distributions from both Droplet Paired-Tag and the Paired-Tag are shown, and the greatest separation between real and random background (*D*) for each dataset is annotated inside the plot.

**d**, boxplot showing Spearman's correlation coefficients of H3K27ac- or H3K27me3- associated distal cCREs-gene pairs intersected at or not at loop anchors identified in snm3C-seq dataset[52]. *P* value, two-tailed Wilcoxon rank-sum test. For all boxplots, hinges were drawn from the 25th to 75th percentiles, with the middle line denoting the median, whiskers with maximum 1.5 interquartile range (IQR). n = 11,231 (H3K27ac, not at loop anchors), 9,010 (H3K27ac, at loop anchors), 2,553 (H3K27me3, not at loop anchors), 2,185 (H3K27me3, at loop anchors). **e**, Representative snm3C-seq contact heatmap and genome browser view showing class-specific (glutamatergic versus non-neurons), active / repressive putative cCREs regulating gene Neurod6. Distal cCREs and proximal regions are highlighted in pink.

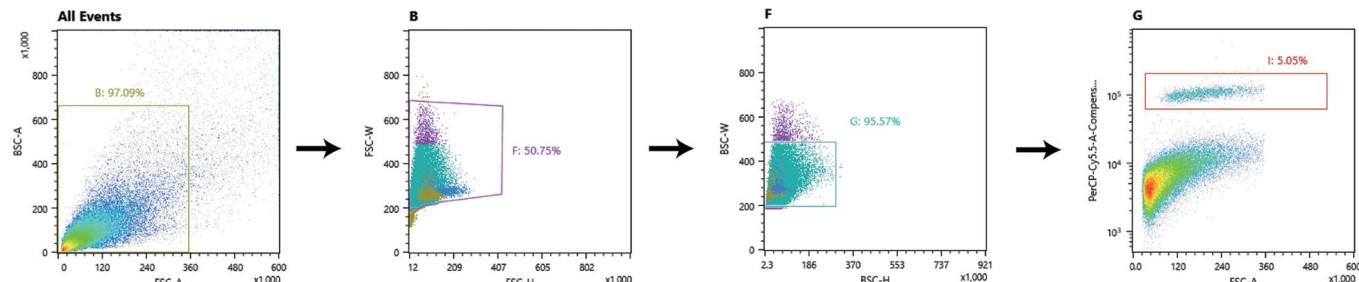

**Extended Data Fig. 9 | Nuclei gating strategy for mouse frontal cortex.**
After nuclei extraction, nuclei were stained with DRAQ7 and proceeded to fluorescence-activated nuclei sorting. First, potential nuclei were identified using forward scatter (FSC) area and backscatter (BSC) area (left dot plot). Next, potential doublets were removed based on BSC as well as FSC signal width (two middle dot plots). Finally, 200–500k diploid nuclei (2n) were collected for antibody incubation (right dot plot).

# Reporting Summary

## Statistics

For all statistical analyses, confirm that the following items are present in the figure legend, table legend, main text, or Methods section.

| n/a | Confirmed | |
|---|---|---|
| ☐ | ☒ | The exact sample size (*n*) for each experimental group/condition, given as a discrete number and unit of measurement |
| ☐ | ☒ | A statement on whether measurements were taken from distinct samples or whether the same sample was measured repeatedly |
| ☐ | ☒ | The statistical test(s) used AND whether they are one- or two-sided *Only common tests should be described solely by name; describe more complex techniques in the Methods section.* |
| ☒ | ☐ | A description of all covariates tested |
| ☐ | ☒ | A description of any assumptions or corrections, such as tests of normality and adjustment for multiple comparisons |
| ☐ | ☒ | A full description of the statistical parameters including central tendency (e.g. means) or other basic estimates (e.g. regression coefficient) AND variation (e.g. standard deviation) or associated estimates of uncertainty (e.g. confidence intervals) |
| ☐ | ☒ | For null hypothesis testing, the test statistic (e.g. *F*, *t*, *r*) with confidence intervals, effect sizes, degrees of freedom and *P* value noted *Give P values as exact values whenever suitable.* |
| ☒ | ☐ | For Bayesian analysis, information on the choice of priors and Markov chain Monte Carlo settings |
| ☒ | ☐ | For hierarchical and complex designs, identification of the appropriate level for tests and full reporting of outcomes |
| ☐ | ☒ | Estimates of effect sizes (e.g. Cohen's *d*, Pearson's *r*), indicating how they were calculated |

*Our web collection on statistics for biologists contains articles on many of the points above.*

## Software and code

Policy information about availability of computer code

| | |
|---|---|
| Data collection | Illumina bcl2fastq2 |
| Data analysis | 10XGenomics cellranger (v2.0.0), 10XGenomics cellranger-atac (v2.0.0), 10XGenomics cellranger-arc (v6.1.2), HOMER (v4.11), MACS2 (v2.1.2), Samtools (v1.14), Seurat (v4.1.0), Signac (v1.6.0), deepTools (v3.5.1), IGV (v.2.15.4), Picard (v2.25.0), Monocle3 (v1.2.7), pairtools (v1.0.2), bedtools (v2.27.1), Juicer tools (v1.22.01), PANTHER (v17.0) Custom scripts and code to reproduce figures are available at: https://github.com/Xieeeee/Droplet-Paired-Tag |

For manuscripts utilizing custom algorithms or software that are central to the research but not yet described in published literature, software must be made available to editors and reviewers. We strongly encourage code deposition in a community repository (e.g. GitHub). See the Nature Portfolio guidelines for submitting code & software for further information.

## Data

Policy information about availability of data

All manuscripts must include a data availability statement. This statement should provide the following information, where applicable:

- Accession codes, unique identifiers, or web links for publicly available datasets
- A description of any restrictions on data availability
- For clinical datasets or third party data, please ensure that the statement adheres to our policy

Raw data obtained in this study have been deposited at the NCBI Gene Expression Omnibus (GEO) (http://www.ncbi.nlm.nih.gov/geo/) with accession number

## Research involving human participants, their data, or biological material

Policy information about studies with human participants or human data. See also policy information about sex, gender (identity/presentation), and sexual orientation and race, ethnicity and racism.

| | |
|---|---|
| Reporting on sex and gender | N/A |
| Reporting on race, ethnicity, or other socially relevant groupings | N/A |
| Population characteristics | N/A |
| Recruitment | N/A |
| Ethics oversight | N/A |

Note that full information on the approval of the study protocol must also be provided in the manuscript.

# Field-specific reporting

Please select the one below that is the best fit for your research. If you are not sure, read the appropriate sections before making your selection.

☒ Life sciences　　☐ Behavioural & social sciences　　☐ Ecological, evolutionary & environmental sciences

For a reference copy of the document with all sections, see nature.com/documents/nr-reporting-summary-flat.pdf

# Life sciences study design

All studies must disclose on these points even when the disclosure is negative.

| | |
|---|---|
| Sample size | Sample size was determined based on prior published data from similar experiments (Preissl et.al., Nat. Neuroscience, 2018; Cao et.al., Science, 2018). To evaluate robustness of the methods, each set of experiment was carried out with tissues samples dissected from three individuals. |
| Data exclusions | Low quality single nuclei (low number of reads / FRiP, low number of genes captured) were excluded from downstream analysis as outlined in the Methods section. Genes (GENCODE vm25) with sufficient levels of transcription (RPKM > 1), or genomic regions with sufficient levels of epigenomic signal (RPKM > 1 for already identified cCREs) were retained for subsequent analysis. |
| Replication | Three biological replicates were performed for each set of experiment. All datasets from independent replicates showed similar results. |
| Randomization | Allocation was random. |
| Blinding | The experiments were not blinded since identities of histone modification targets and tissue regions are needed to evaluate the specificity and sensitivity of the method. Clustering of single-nuclei transcriptome and epigenome data were unsupervised. |

# Reporting for specific materials, systems and methods

We require information from authors about some types of materials, experimental systems and methods used in many studies. Here, indicate whether each material, system or method listed is relevant to your study. If you are not sure if a list item applies to your research, read the appropriate section before selecting a response.

## Materials & experimental systems

| n/a | Involved in the study |
|-----|----------------------|
| ☐ | ☒ Antibodies |
| ☐ | ☒ Eukaryotic cell lines |
| ☒ | ☐ Palaeontology and archaeology |
| ☐ | ☒ Animals and other organisms |
| ☒ | ☐ Clinical data |
| ☒ | ☐ Dual use research of concern |
| ☒ | ☐ Plants |

## Methods

| n/a | Involved in the study |
|-----|----------------------|
| ☒ | ☐ ChIP-seq |
| ☐ | ☒ Flow cytometry |
| ☒ | ☐ MRI-based neuroimaging |

# Antibodies

| | |
|---|---|
| Antibodies used | Antibodies used in this study include: H3K27ac (Abcam, ab177178, Lot GR3202987-20 (recombinant); Abcam, ab4729, Lot GR3442886-1 (polyclonal)) and H3K27me3 (Abcam, ab192985, Lot GR3399022-3 (recombinant)). For all assays, 1 µg of antibody was used for 200 - 500k nuclei, in 75 µL reaction system. |
| Validation | All antibodies used in this study are commercially available, and have been validated appropriate for the assays used in this manuscript by the provider (https://www.abcam.com/products/primary-antibodies/histone-h3-acetyl-k27-antibody-ep16602-chip-grade-ab177178.html; https://www.abcam.com/products/primary-antibodies/histone-h3-acetyl-k27-antibody-chip-grade-ab4729.html; https://www.abcam.com/products/primary-antibodies/histone-h3-tri-methyl-k27-antibody-epr18607-chip-grade-ab192985.html). <br> We also validated all antibodies with in-house CUT&Tag and compared to ENCODE data(H3K27ac (ENCSR000AOC), H3K27me3 (ENCSR000DTY)) before using for publication. <br> We found that antibodies specificity is critical for high-quality signals of single-cell histone data. For H3K27ac, although recombinant antibody yielded a higher fragment number per cell than polyclonal antibodies, its enrichment at transcription starting sites or ChIP-seq peaks was lower. Therefore, except for replicate 1 (rep1) of the mouse frontal cortex datasets, all other experiments targeting in H3K27ac were carried out with the polyclonal antibody. |

# Eukaryotic cell lines

Policy information about cell lines and Sex and Gender in Research

| | |
|---|---|
| Cell line source(s) | Mouse embryonic stem cells (mESC) used in this study is genetically engineered to have allelic tagging of Sox2 genes with egfp and mcherry on CAST and 129/Sv allele, respectively, and also harbors an insertion of four CTCF binding sites in between Sox2 and its downstream super-enhancer on the CAST allele (Huang et.al., 2021, Nat. Genetics). The parental, hybrid F123 mESC line (F1 Mus musculus castaneus×S129/SvJae, maternal 129/Sv, paternal CAST) was from Rudolf Jaenisch's laboratory at the Whitehead Institute at MIT. |
| Authentication | Cells were not authenticated. |
| Mycoplasma contamination | Cells were not tested for mycoplasma. |
| Commonly misidentified lines (See ICLAC register) | None of the cell lines used are listed in the ICLAC database. |

# Animals and other research organisms

Policy information about studies involving animals; ARRIVE guidelines recommended for reporting animal research, and Sex and Gender in Research

| | |
|---|---|
| Laboratory animals | All animal work described in this manuscript has been approved and conducted under the oversight of the UC San Diego Institutional Animal Care and Use Committee. Male C57BL/6J mice were purchased from the Jackson Laboratory (#000664) at 12 weeks of age and housed in the barrier facility at UC San Diego in a 12-hour light/dark cycle in a temperature-controlled room, with ad libitum access to water and food, until euthanasia and tissue collection at 16 weeks of age. The temperature in the animal facility is maintained within the range of 68°F to 72°F, while the humidity levels can vary between 35% and 60%. The frontal cortex was dissected from 16-week male mice, snap-frozen in liquid nitrogen, and stored at -80°C before proceeding to nuclei extraction. |
| Wild animals | The study did not involve wild animals. |
| Reporting on sex | All three replicates used in this study are male C57BL/6J mice |
| Field-collected samples | The study did not involve field collected samples |
| Ethics oversight | All animal work described in this manuscript has been approved and conducted under the oversight of the UC San Diego Institutional Animal Care and Use Committee. |

Note that full information on the approval of the study protocol must also be provided in the manuscript.

# Flow Cytometry

## Plots

Confirm that:

☒ The axis labels state the marker and fluorochrome used (e.g. CD4-FITC).

☒ The axis scales are clearly visible. Include numbers along axes only for bottom left plot of group (a 'group' is an analysis of identical markers).

☒ All plots are contour plots with outliers or pseudocolor plots.

☒ A numerical value for number of cells or percentage (with statistics) is provided.

## Methodology

| | |
|---|---|
| Sample preparation | Single cell suspensions were prepared from frozen mouse cortex by douncing in the douncing buffer (0.25 M sucrose (Sigma, S7903), 25 mM KCl (Sigma, P9333), 5 mM MgCl2, 10mM Tris-HCl pH 7.4, 1 mM DTT (Sigma, D9779), 1× Protease Inhibitor, 0.5 U µl−1 Rnase OUT, 0.5 U µl−1 SUPERase Inhibitor). The cell suspension was then filtered through a 30-µm Cell-Tric (Sysmex) for debris removal, and spun down for 10min at 300g, 4°C. Cell pellets were washed once with douncing buffer, spun down again, and resuspended in cold nuclei permeabilization buffer for 10mins. Permeabilized nuclei were pelleted by centrifuge for 10min at 1,000g, 4°C, and washed with sort buffer (1× PBS (Gibco, 10010023), 1×Protease Inhibitor (Roche, 05056489001), 0.5 U µl−1 Rnase OUT (Invitrogen, 10777-019), 0.5 U µl−1 SUPERase Inhibitor (Invitrogen, AM2694), 1mM EDTA (Invitrogen, 15575020), 1% BSA (Sigma, A1595)) once. After resuspension in sort buffer, nuclei are stained with 2 µM 7-AAD (Invitrogen, A1310) for 10mins on ice, and proceed to Fluorescence-activated cell sorting (FACS) with an SH800 cell sorter (Sony) for isolation of single nucleus. |
| Instrument | SH800 Cell Sorter (Sony) |
| Software | Data analysis and display are performed in the SH800 software. |
| Cell population abundance | We used 7-AAD to stain intact nuclei and separated them from debris, without separating identities of different cell types. Using 'normal' mode in sorting nuclei, we collected 200,000-500,000 nuclei for each sample for down-streaming processing. |
| Gating strategy | First, potential nuclei were identified using forward scatter (FSC) area and back scatter (BSC) area. Next, potential doublets were removed based on BSC and FSC signal width. Finally, diploid nuclei (2n) were sorted into each tube for down-streaming processing. |

☒ Tick this box to confirm that a figure exemplifying the gating strategy is provided in the Supplementary Information.

