## [Peer Review File · Nature Structural & Molecular Biology]

Peer Review Information

Manuscript Title: Droplet-based Single-cell Joint Profiling of Histone Modification and Transcriptome

Corresponding author name(s): Bing Ren

Reviewer Comments & Decisions:

Decision Letter, initial version:

Message: 15th May 2023

Dear Prof. Ren,

Thank you again for submitting your manuscript "Droplet-based Single-cell Joint Profiling of Histone Modification and Transcriptome". I apologise for the delay in responding, which resulted from the difficulty in obtaining suitable referee reports. Nevertheless, we now have comments (below) from the 3 reviewers who evaluated your paper. In light of those reports, we remain quite interested in your study and would like to see your response to the comments of the referees, in the form of a revised manuscript.

You will see that all referees appreciate the progress achieved in making the technique more accessible to a broader audience and the benchmarking carried out to validate its applicability. There are, however, a few important issues that should be addressed in a revision. First and foremost, we ask that you fix the github link containing the accompanying code, as also mentioned by the experts. Secondly, we request that you follow the guidelines of the reviewers with respect to performing some additional analyses and benchmarking, textually clarifying ambiguities where indicated, expanding sections to include additional details that the experts deem will be useful to the readers/users, and moving some panels to highlight important parts of the story.

Please be sure to address/respond to all concerns of the referees in full in a point-by-point response and highlight all changes in the revised manuscript text file. If you have comments that are intended for editors only, please include those in a separate cover letter.

We are committed to providing a fair and constructive peer-review process. Do not hesitate to contact us if there are specific requests from the reviewers that you believe are

technically impossible or unlikely to yield a meaningful outcome.

We expect to see your revised manuscript within 2-3 months. If you cannot send it within this time, please contact us to discuss an extension; we would still consider your revision, provided that no similar work has been accepted for publication at NSMB or published elsewhere.

Reporting Summary:

When submitting the revised version of your manuscript, please pay close attention to our [href="https://www.nature.com/nature-portfolio/editorial-policies/image-integrity">Digital Image Integrity Guidelines. and to the following points below:](https://www.nature.com/nature-portfolio/editorial-policies/image-integrity)

Data availability: this journal strongly supports public availability of data. All data used in accepted papers should be available via a public data repository, or alternatively, as Supplementary Information. If data can only be shared on request, please explain why in your Data Availability Statement, and also in the correspondence with your editor. Please

note that for some data types, deposition in a public repository is mandatory - more information on our data deposition policies and available repositories can be found below: <https://www.nature.com/nature-research/editorial-policies/reporting-standards#availability-of-data>

[Redacted]

Sincerely,

Dimitris Typas
Associate Editor
Nature Structural & Molecular Biology
ORCID: 0000-0002-8737-1319

Referee expertise:

Referee #1: Single-cell omics in brain

Referee #2: Single-cell (epi)genomics, technique development

Referee #3: Single-cell omics, technique development

Reviewers' Comments:

Reviewer #1:

Remarks to the Author:

In the manuscript "Droplet-based Single-cell Joint Profiling of Histone Modification and Transcriptome", Xie, Zhu and colleagues report a droplet-based iteration of their previous method Paired-Tag, that allows the simultaneous profiling of the transcriptome and the deposition of specific histone modifications genome-wide in thousands of single cells. With the development of droplet-based multiomic RNA/ATAC approaches by 10x Genomics and droplet-base single-cell methods for CUT&Tag also based on the 10x platform, the combination of these methods is an expected development that many researchers are looking forward to use. The protocol described by the authors achieves this goal and will allow the fast implementation of this technology by the research community, given the widespread use of the 10x platform. Thereby, I recommend an expedited publication of the paper in NSMB, pending on addressing some minor points:

- In order to make the method more understandable for readers not familiar with these technologies, the authors should expand Figure 1a and give more detail for instance on the part of the protocol occurring after the ligation and reverse transcription. Also, in Extended Figure 2, the full scheme of oligos and primers (including sequences) should be included.
- Related with the previous point, in sentence 70-71, the authors mention "Reverse transcription and template switching reactions are subsequently carried out in the droplets, and tagmented chromatin fragments are ligated to DNA oligos with unique molecular barcode sequences using a bridge adaptor". The authors should explain in more detail the concept of bridge adaptor, for instance referring that the adaptor is actually added via the initial CUT&Tag tagmentation.
- The authors use as starting material 0,5 million cells, of which 10 000 – 16 000 cells are used to load in the 10x instrument. Is there a considerable loss of cells during the protocol? The authors should comment on this aspect and compare their method to other methods, namely what would be the minimum number of cells required, since this aspect will be critical for the implementation of the technology.
- Figure 1h – clarify if the y-axis bulk refers to ChIP or CUT&Tag
- Figure 2a
 - o How does the data compare with other single cell epigenomic datasets using mouse brain, as for instance Bartosovic et al., Nature Biotechnology 2022? Could the authors integrate the datasets and confirm that the identify of the clusters match those of Bartosovic et al., even if they represent different developmental stages?

o The authors comment that there is a higher number of clusters obtained by the RNA analysis, but could this be due to different numbers of cells analysed? Do the authors obtain similar results if the clustering is performed for the RNA part in the individual experiments, having similar number of cells to the CUT&Tag part of the experiments?

o There seems to be a higher number of OPCs in the H3K27ac and H3K27me3 UMAPs, but this might be due to the scaling. It would be more representative to present UMAPs with a downscaling of the number of cells for RNA to the same level of the histone marks

- The authors mention in the methods ("Antibody") that there are differences between the H3K27ac antibodies and that they use different antibodies in different replicates. These differences in number of fragments per cell and FRIP should be shown in the Extended Figures.

- Figure 2e, f – "The authors mentions in sentence 138 : "For both histone modifications profiled, Droplet Paired-Tag yielded a comparable or higher fraction of reads in peaks than a previously reported scCUT&Tag study and the combinatorial-indexing based Paired-Tag datasets, which could explain the higher resolution in separating cell types." However, it is clear from the figures that the number of unique fragments per cells is much higher in standard Paired-Tag compared to droplet Paired-Tag. The authors should rephrase and comment on why this might be the case.

- Figure 2h- Can the authors also show the overlap between droplet Paired-tag and standard Paired-tag?

- I would recommend moving Extended Figure 5c in the main figure, since it illustrate the power of the technology in a better way then Figures 2i-m

- The github code link is not operational.

Reviewer #2:

Remarks to the Author:

The manuscript "Droplet-based Single-cell Joint Profiling of Histone Modification and Transcriptome" focuses on technical implementations for a multiomic single-cell profiling method, Paired-Tag. Paired-Tag was first described by this group a couple of years ago, and performs RNA-seq transcriptome profiling and CUT&Tag chromatin profiling. The original implementation uses 96 -well plates, and the major accomplishment in the current manuscript is moving this method onto a commercial 10X droplet machine. This workflow is substantially faster and, as using a commercial platform, will be more available and reliable for other labs. The data presented here verifying the technique is nicely demonstrated and straightforward.

Reviewer #3:

Remarks to the Author:

Xie et al. report Droplet Paired-Tag, an adaptation of the existing Paired-Tag method previously reported by the same group (Zhu et al. 2021). This adaptation improves upon Paired-Tag by making it compatible with widely used droplet-based single-cell isolation

platforms that are more user-friendly than the combinatorial indexing-based approach upon which the original Paired-Tag method was based. The authors show that droplet-based Paired-Tag does essentially everything that Paired-Tag can do (including cell type identification in mouse frontal cortex, comparison of gene expression and histone modification enrichment in the same cells, identification of candidate regulatory elements, and prediction of functional regulatory factors via motif analysis), with similar or improved fidelity and resolution. This short report will be of utility to the field by furnishing a method that stands a better chance of being adopted and deployed by non-expert users. However, the authors would do well to clarify some analysis points, including the details of cluster definition that form the basis of quality comparisons between droplet Paired-Tag and other similar methods. They also must make the code available for review—the link provided was broken.

Points of critique:

- Figure 1h: How does the saturation of peaks compare with combinatorial indexing Paired-Tag, scCUT&Tag, and/or other similar methods?
- Figure 2a-b: The authors state that "...out of the 20 major cell clusters, 17 and 18 can also be recovered by independent clustering using histone modification H3K27ac or H3K27me3 signals, respectively...". Can the authors expand on how the clusters were defined? For instance:
 - o For the sake of reproducibility, can the authors report the relevant details of RNA and epigenome cluster definition via Seurat/Signac, including which SVD dimensions used for UMAP projection for each modality? Are the clusters reported here "recovered" under the same exact dimensionality reduction and clustering conditions? I assume this information can be found in the code, but when I attempted to visit the Github link it appeared to be broken.
 - o The methods indicate that marker genes recovered from previous studies were used for cluster annotation. What was the full complement of specific marker genes used to define each cluster? What data support their use as markers/what were the previous studies used?
 - o Can the authors assign a confidence value to cluster annotation in comparison with marker gene enrichment in clusters in the previous studies that were used to define them? Was there an in-cluster vs. out-of-cluster marker gene enrichment threshold for which a cluster was considered to have been "recovered"? Extended Data Figure 4c-e indicates that clusters defined here are similar in enrichment to similarly defined clusters from previous studies, but this is largely circular logic since presumably these cluster annotations were defined the same way across all studies, so it doesn't speak to the degree to which the marker genes themselves are uniquely enriched in clusters in any of these studies.
 - o How was H3K27me3 treated for cluster discovery? By the absence of H3K27me3 at in-cluster marker genes alone, or by the presence of H3K27me3 in out-of-cluster marker genes as well? Given the sparsity of the data, the latter approach seems more prudent.
- As noted above, the Github link to the code

Author Rebuttal to Initial comments

Point-by-point Response to Reviewer #1

“In the manuscript “Droplet-based Single-cell Joint Profiling of Histone Modification and Transcriptome”, Xie, Zhu and colleagues report a droplet-based iteration of their previous method Paired-Tag, that allows the simultaneous profiling of the transcriptome and the deposition of specific histone modifications genome-wide in thousands of single cells. With the development of droplet-based multiomic RNA/ATAC approaches by 10x Genomics and droplet-base single-cell methods for CUT&Tag also based on the 10x platform, the combination of these methods is an expected development that many researchers are looking forward to use. The protocol described by the authors achieves this goal and will allow the fast implementation of this technology by the research community, given the widespread use of the 10x platform. Thereby, I recommend an expedited publication of the paper in NSMB, pending on addressing some minor points:”

Response: We thank the referee very much for the positive comments of our method and constructive suggestions!

1. “In order to make the method more understandable for readers not familiar with these technologies, the authors should expand Figure 1a and give more detail for instance on the part of the protocol occurring after the ligation and reverse transcription. Also, in Extended Figure 2, the full scheme of oligos and primers (including sequences) should be included.”

Response: Thank you for this suggestion! In the revised manuscript, we have revised Fig. 1a and Extended Data Fig. 2a to explain how DNA and cDNA are barcoded and amplified for construction of the sequencing libraries in the Droplet Paired-Tag procedure. We also added a Supplementary Table 1 to provide the oligo sequences used in this study.

2. “Related with the previous point, in sentence 70-71, the authors mention “Reverse transcription and template switching reactions are subsequently carried out in the droplets, and tagmented chromatin fragments are ligated to DNA oligos with unique molecular barcode sequences using a bridge adaptor”. The authors should explain in more detail the concept of bridge adaptor, for instance referring that the adaptor is actually added via the initial CUT&Tag tagmentation.”

Response: The barcoded beads contain two types of barcoded oligos: an oligo containing poly-dT for cDNA synthesis and an oligo containing capturing sequencing for labeling of tagmented DNA fragments. In the droplets, reverse transcription is carried out with the barcoded poly-dT oligo. A ligation reaction is simultaneously performed to attach the second barcoded oligo to tagmented chromatin fragments. We designed the pA-Tn5 adaptor with a 3'-extended bridge-pMENTs sequence that is reverse complementary with the second capturing oligo to facilitate the tagmented chromatin ligation reaction. Upon completion of both reactions, the cDNA and tagmented chromatin fragments from the same cells are labeled with the same cellular barcodes and amplified for the construction of the sequencing libraries. We have revised this part (Lines 69-82) to increase clarity. Thank you for this great suggestion!

3. “The authors use as starting material 0,5 million cells, of which 10 000 – 16 000 cells are used to load in the 10x instrument. Is there a considerable loss of cells during the protocol? The authors should comment on this aspect and compare their method to other methods, namely what would be the minimum number of cells required, since this aspect will be critical for the implementation of the technology.”

Response: We have performed systematic optimization on the amount of starting materials and found the optimal numbers of starting cells number ranging from 0.15 -

0.50 million. Multiple factors are considered: (1) A minimum number of cells is required for multiple centrifugation-based washing steps during immunostaining of nuclei - we cannot adopt ConA beads-based washing (as in bulk CUT&Tag (Kaya-Okur, *et al.*, 2019) with lower input requirements) is because the beads bound nuclei cannot be encapsulated into liquid droplets. (2) The maximum number of nuclei is determined considering the total amount of antibody and pA-Tn5 required for optimal tagmentation reaction. (3) The previous Paired-Tag procedure (Zhu *et al.*, 2021) requires 3.6 million cells for ligation-based combinatorial barcoding, which is multiplexed from 12 individual immunostaining reactions (of 0.3 million nuclei). Using the droplet platform, we can either perform 1 reaction of 0.15 million cells or parallelly perform multiple reactions at a time. Our minimal input number is also similar to the requirement of scCUT&Tag (Bartosovic *et al.*, 2021); (4) According to 10X Genomics Multiome protocol, only 5 - 20k nuclei can be loaded to one channel of the droplet generation system. We normally recover 30 - 60% of the starting nuclei right before loading the microfluidics; therefore, we can either load one sample for multiple channels or load different samples for different channels depending on the number of nuclei required for the analysis. To increase clarity, we have included this information in the revised manuscript (Lines 304-309). Thanks for the suggestion!

4. “Figure 1h – clarify if the y-axis bulk refers to ChIP or CUT&Tag”

Response: We thank the reviewer for pointing this out. In the previous manuscript, the y-axis refers to the peaks from pseudobulk Droplet Paired-Tag dataset. In this revised manuscript, we include a comparison of peak saturation to the combinatorial-indexing Paired-Tag dataset, as suggested by reviewer #3. Therefore, the current y-axis refers to the peaks recovered from the corresponding Droplet Paired-Tag / Paired-Tag pseudobulk datasets.

5. “Figure 2a: How does the data compare with other single cell epigenomic datasets using mouse brain, as for instance Bartosovic et al., Nature Biotechnology 2022? Could the authors integrate the datasets and confirm that the identify of the clusters match those of Bartosovic et al., even if they represent different developmental stages?”

Response: We agree with this suggestion. In this revision, we performed the cross-datasets integration with conventional Paired-Tag (Zhu *et al.*, 2021), scCUT&Tag (Bartosovic *et al.*, 2021) and nano-CT (Bartosovic *et al.*, 2022) and compared the cell clustering results. For nano-CT, since the single modality H3K27ac data is not available from GEO, we excluded it in the integration.

In summary, for H3K27ac, the non-neuron cell types match well between all datasets, while for dataset using adult frontal cortex (Paired-Tag and Droplet Paired-Tag), the identities of different neuronal cell types are also concordant, except for cells type that are originated from different brain regions (e.g., CA, DG and Subiculum neurons from Paired-Tag, OBGA and D12MSN from Droplet Paired-Tag), consistent with the RNA integration result in the revised Extended Fig. 5e-g.

For H3K27me3, the classification of non-neurons or neurons class matched well between Droplet Paired-Tag and scCUT&Tag. However, the overlap between Droplet Paired-Tag with Paired-Tag and nano-CT is lower, which can be explained by several factors: (1) The clustering results of Paired-Tag H3K27me3 fail to resolve major cell types, as noted in the original publication (Zhu *et al.*, 2021). (2) Mouse brain samples used by nano-CT were dissected from post-natal day 19 mice, where neurons, astrocytes and oligodendrocytes maturation are still ongoing. (3) Mouse brain samples

used by nano-CT are whole brain, where the cell type composition could be substantially different from those of adult frontal cortex.

These integration results are now included in the revised Extended Fig. 6f-i. To keep the consistency of data used for comparison in the manuscript, we only displayed the integration results for Droplet Paired-Tag, Paired-Tag, and scCUT&Tag.

6. “The authors comment that there is a higher number of clusters obtained by the RNA analysis, but could this be due to different numbers of cells analysed? Do the authors

obtain similar results if the clustering is performed for the RNA part in the individual experiments, having similar number of cells to the CUT&Tag part of the experiments?”

Response: We agree with the reviewer that the number of cells could influence the final clustering results. In the initial analysis, three cell groups identified by RNA (with ~22k nuclei) could not be resolved from histone datasets without subclustering, including CLAGL (Glutamatergic neurons, Claustrum), VLMC (Vascular and leptomeningeal cells) and STRGA (GABAergic neurons, Striatum). To rule out the possibility that this difference in cell clustering between RNA and histone signal is due to cell numbers we re-did the cell clustering analysis after downsampling the datasets. Specifically, we downsampled the total cell number to 15,000, 10,000 (which is close to the number of H3K27ac / H3K27me3 cells), 5,000, 1,000, 500, and 200. With different cells number, we calculated the overlap score between unsupervised Louvain clustering (resolution = 1) and cell type labels inherited from the complete dataset (22,054 nuclei): (1) with 10,000 nuclei, the RNA-based clustering could resolve similar cell type identification as in the 22-k nuclei analysis; (2) with 5,000 nuclei (~50% of nuclei compared to the histone datasets), CLAGL was found to be overlapped with cell type ITL45GL (Glutamatergic neurons, Cortex L4 / L5); (3) when we further downsampled to 500 nuclei, VLMC merged with MGL (Microglia), and STRGA mix with D12MSN (D1 / D2 like medium spiny neurons, Striatum) population, similar as the observation from 10k nuclei H3K27ac-based clustering, possibly due to the skewed selection of variable genes when cell number is low. These results suggested that when the cell number is low (less than 5,000), increasing the cell number could improve cell clustering. When the cell number is higher, RNA-based clustering has a similar resolution between 10k and 20k nuclei and outperforms H3K27ac-based cell clustering even with 50% of the cell numbers. To increase clarity, we have added these new results to the revised Extended Data Fig. 4d. Thanks for the suggestion!

7. “There seems to be a higher number of OPCs in the H3K27ac and H3K27me3 UMAPs, but this might be due to the scaling. It would be more representative to present UMAPs with a downscaling of the number of cells for RNA to the same level of the histone marks.”

Response: We agree that the representation of cell numbers in different populations can be skewed by UMAP plotting. Per the request, we have added the panels showing RNA modality UMAPs with the same numbers of nuclei. We also added the pie charts showing the numbers of nuclei of different cell types in each dataset (the revised Extended Data Fig. 5a-b).

8. “The authors mention in the methods (“Antibody”) that there are differences between the H3K27ac antibodies and that they use different antibodies in different replicates. These differences in number of fragments per cell and FRIP should be shown in the Extended Figures.”

Response: We thank the reviewer for pointing this out. We have added the unique fragments per cell and FRiP for datasets generated with different H3K27ac antibodies in the revised Extended Data Fig. 6c.

9. “Figure 2e, f – “The authors mentions in sentence 138 : “For both histone modifications profiled, Droplet Paired-Tag yielded a comparable or higher fraction of reads in peaks than a previously reported scCUT&Tag study and the combinatorial-indexing based Paired-Tag datasets, which could explain the higher resolution in separating cell types.” However, it is clear from the figures that the number of unique fragments per cells is much higher in standard Paired-Tag compared to droplet Paired-Tag. The authors should rephrase and comment on why this might be the case.”

Response: We apologize for the confusion. We used two metrics to evaluate the performance of CUT&Tag modality, including (1) the number of unique fragments per cell and (2) the fraction of reads in peaks (FRiP). Compared to scCUT&Tag on the 10X platform, we yielded comparable FRiP but higher median numbers of unique fragments per nuclei. Compared to conventional Paired-Tag, although fewer reads from each cell were captured, the FRiP is higher and thus gives comparable/or higher peak-associated reads. The higher cell clustering results are contributed by the improvements in both signal sensitivity (numbers of unique fragments per nuclei) and specificity (FRiP). We have revised these statements in Lines 146-151 of the manuscript. Thank you for pointing this out!

10. “Figure 2h- Can the authors also show the overlap between droplet Paired-tag and standard Paired-tag?”

Response: Per the request, we have performed a comparison of peak calling from Droplet and standard Paired-Tag: by downsampling the numbers of reads into the same level for H3K27ac dataset, we called 27,522 and 63,799 peaks from standard and Droplet Paired-Tag, respectively; among them, 14,288 (51.9% of peaks called from standard Paired-Tag dataset) overlapped with each other. We have added the new results into Extended Data Fig. 6d. Thanks for the suggestion!

11. “I would recommend moving Extended Figure 5c in the main figure, since it illustrate the power of the technology in a better way then Figures 2i-m”

Response:

We agree with the reviewer that panels i-k is a bit redundant with panels l-m as they are examples of how Paired-Tag datasets could be used for the identification of cell type-specific proximal or distal cCREs and for prediction of the associated TFs. The previous Extended Data Fig. 5e showed the relationships between densities of cCRE histone modifications and the expression levels of their predicted target genes. We have moved this panel to the main figure to replace the original panel Fig. 2l-m. Thank you for the suggestion!

12. “The github code link is not operational.”

Response: We apologize for the broken GitHub link. We have fixed it now.

Point-by-point Response to Reviewer #2

“The manuscript “Droplet-based Single-cell Joint Profiling of Histone Modification and Transcriptome” focuses on technical implementations for a multiomic single-cell profiling method, Paired-Tag. Paired-Tag was first described by this group a couple of years ago, and performs RNA-seq transcriptome profiling and CUT&Tag chromatin profiling. The original implementation uses 96 -well plates, and the major accomplishment in the current manuscript is moving this method onto a commercial 10X droplet machine. This workflow is substantially faster and, as using a commercial platform, will be more available and reliable for other labs. The data presented here verifying the technique is nicely demonstrated and straightforward.”

Response: We thank this referee for the positive comments!

Point-by-point Response to Reviewer #3

“Xie et al. report Droplet Paired-Tag, an adaptation of the existing Paired-Tag method previously reported by the same group (Zhu et al. 2021). This adaptation improves upon Paired-Tag by making it compatible with widely used droplet-based single-cell isolation platforms that are more user-friendly than the combinatorial indexing-based approach upon which the original Paired-Tag method was based. The authors show that droplet-based Paired-Tag does essentially everything that Paired-Tag can do (including cell type identification in mouse frontal cortex, comparison of gene expression and histone modification enrichment in the same cells, identification of candidate regulatory elements, and prediction of functional regulatory factors via motif analysis), with similar or improved fidelity and resolution. This short report will be of utility to the field by furnishing a method that stands a better chance of being adopted and deployed by non-expert users. However, the authors would do well to clarify some analysis points, including the details of cluster definition that form the basis of quality comparisons between droplet Paired-Tag and other similar methods. They also must make the code available for review—the link provided was broken.”

Response: We thank this referee for the positive comments and suggestions! We have made the suggested revisions. Please find the detailed response below.

1. “Figure 1h: How does the saturation of peaks compare with combinatorial indexing Paired-Tag, scCUT&Tag, and/or other similar methods?”

Response: As suggested, we have modified the Extended Fig. 3e to include the peak saturation analysis by reads comparing Droplet Paired-Tag and combinatorial indexing Paired-Tag data on the same mESC cell line. Since the sequencing coverage and duplication rate per cell varies between two datasets, we didn’t compare the peaks

saturation analysis by cells. Compared with Droplet Paired-Tag, Paired-Tag requires 60% more reads to reach peak saturation (defined by recovery of 85% of peaks identified from the pseudobulk dataset).

2. “Figure 2a-b: The authors state that “...out of the 20 major cell clusters, 17 and 18 can also be recovered by independent clustering using histone modification H3K27ac or H3K27me3 signals, respectively...”. Can the authors expand on how the clusters were defined? For instance:

o For the sake of reproducibility, can the authors report the relevant details of RNA and epigenome cluster definition via Seurat/Signac, including which SVD dimensions used for UMAP projection for each modality? Are the clusters reported here “recovered” under the same exact dimensionality reduction and clustering conditions? I assume this information can be found in the code, but when I attempted to visit the Github link it appeared to be broken.

Response: We apologize for the broken GitHub link, which has now been fixed. As requested, we added the relevant details to the method section (Clustering and annotation), including the number of variable features (genes / genomic bins) and principle components selected for clustering. To assist readers to reproduce our results, we have uploaded the full codes for generating the clustering results in the Jupyter Notebook format on GitHub (<https://github.com/Xieeeee/Droplet-Paired-Tag/blob/main/02.analysis/notebook/02.Clustering.ipynb>).

o The methods indicate that marker genes recovered from previous studies were used for cluster annotation. What was the full complement of specific marker genes used to define each cluster? What data support their use as markers/what were the previous studies used?

Response: As requested, we have added (1) Supplementary Table 3 which includes the lists of all differentially expressed genes detected for each cell group; (2) Supplementary Table 4 which includes the marker genes used for cell type annotation and the supporting references. Thank you for the suggestion!

o Can the authors assign a confidence value to cluster annotation in comparison with marker gene enrichment in clusters in the previous studies that were used to define them? Was there an in-cluster vs. out-of-cluster marker gene enrichment threshold for which a cluster was considered to have been “recovered”? Extended Data Figure 4c-e indicates that clusters defined here are similar in enrichment to similarly defined clusters from previous studies, but this is largely circular logic since presumably these cluster annotations were defined the same way across all studies, so it doesn’t speak to the degree to which the marker genes themselves are uniquely enriched in clusters in any of these studies.

Response: To annotate cell type identity, we first performed unsupervised Louvain clustering (resolution = 0.8) on the transcriptome data, followed by marker genes identification and then annotated according to the enrichment of known marker genes. To estimate the accuracy of cell type clustering and annotation we carried out co-embedding with the golden standard BICCN 10X snRNA-seq dataset from Allen Institute (Yao *et al.*, 2021) and > 93% of the cells types originated from frontal cortex matched with this reference dataset (revised Extended Fig. 5e-g).

To show the specificity of marker gene expression in different cell groups, we plotted the average expression levels of the representative marker genes used for annotation across all cell groups (revised Extended Data Fig. 4a). As also shown in the plots

below, most of our marker genes used for annotation are specifically enriched in only one or few closely related cell populations. For populations sharing the same marker gene plot here, additional markers are used to further distinguish their identities. The full list of marker genes used for annotation can be found in Supplementary Table 4.

As suggested, we also derived an enrichment score of the marker genes for each cluster. First, we calculated the Log_2 fold enrichment ratio of the normalized gene expression (RPKM) in the cluster of interest versus all other cells in the whole dataset. Next, the fold enrichment ratio is averaged across all marker genes used for cluster classification, which yields our enrichment score. As shown below, more than 98% of clusters showed an enrichment score >1 (average fold change >2), and the average enrichment score across clusters is 4.64. These results suggested that the marker genes we used for annotation are specific and uniquely enriched in the cell populations

we identified. We have included these new results in the revised Extended Data Fig. 4c. Thanks for the suggestion!

o How was H3K27me3 treated for cluster discovery? By the absence of H3K27me3 at in-cluster marker genes alone, or by the presence of H3K27me3 in out-of-cluster marker genes as well? Given the sparsity of the data, the latter approach seems more prudent.

Response: For cell clustering based on H3K27me3, we converted the histone signal into 5kb non-overlapping genomic bins and clustered using the TF-IDF followed by dimension reduction with SVD with the Signac software. As shown in Fig. 2a-b, the cell clustering results show high agreement between RNA and H3K27me3-based analysis, indicating that the presence of H3K27me3 in out-of-cluster marker genes contributed to restricting the cell type identity. During the revision, we analyzed the top 10 cell-type specific H3K27me3 bins from each cluster with >100 cells (VLMC excluded) and found that 87% of these bins overlapped with genes body: the expression levels genes overlapped with H3K27me3 enriched regions showed negative correlations with the H3K27me3 signal. Within these genes, 58.8% are found to be differentially expressed

across different cell types. Thus, the H3K27me3 contributes to cell type identification by repressing out-of-cluster gene expressions. While identifying cell types and states based on H3K27me3 signal alone is challenging, the paired RNA profiles could help to identify the highly expressed marker genes to assist cell type annotation. We have added the new results in the revised Extended Data Fig. 8c-d. Thanks for the suggestion!

- As noted above, the Github link to the cod

Response: We apologize for the broken GitHub link. This issue has been fixed now.

Decision Letter, first revision:

Message: Our ref: NSMB-BC47527A

22nd Jun 2023

Dear Professor Ren,

Thank you for submitting your revised manuscript "Droplet-based Single-cell Joint Profiling of Histone Modification and Transcriptome" (NSMB-BC47527A). It has now been seen by the original referees and their comments are below. The reviewers find that the paper has further improved in revision, and therefore we'll be happy in principle to publish it in Nature Structural & Molecular Biology, pending minor revisions to satisfy the final request of reviewer #3, who notes a discrepancy between panel 2a and the raw data, and to comply with our editorial and formatting guidelines.

Sincerely,

Dimitris Typas
Associate Editor
Nature Structural & Molecular Biology
ORCID: 0000-0002-8737-1319

Reviewer #1 (Remarks to the Author):

The authors addressed my points and I recommend the publication of the manuscript.

Reviewer #2 (Remarks to the Author):

I had no concerns with the first submission of this manuscript. This revision looks good.

Reviewer #3 (Remarks to the Author):

The authors have satisfied nearly all of the critiques/questions I posed in the original review. In particular, I appreciate that the code base has been provided via Jupyter notebook. However, there appears to be a discrepancy between the Figures (specifically Figure 2a) and the notebook: Although the RNA UMAP presented in this Figure appears to be identical the projection generated in the Jupyter notebook, it appears that the two

chromatin UMAPs in Figure 2a are different than what was generated in the notebook. Were the cluster/cell type definition analyses performed based on the UMAPs presented in Figure 2a, or the ones in the Jupyter notebook? The authors should ensure the analysis and projections are fully consistent with each other, lest the conclusions be altered based on the projection used. Moreover, if the cluster analyses change materially based on the projection used, this might undermine the conclusions and should be remedied/noted in some way.

Author Rebuttal, first revision:

Response to reviewers' comments:

Reviewer #1 (Remarks to the Author):

The authors addressed my points and I recommend the publication of the manuscript.

Response: We sincerely appreciate the input provided by the reviewers, which greatly contributed to the enhancement of our manuscript.

Reviewer #2 (Remarks to the Author):

I had no concerns with the first submission of this manuscript. This revision looks good.

Response: We really appreciate the positive feedbacks from the reviewers.

Reviewer #3 (Remarks to the Author):

The authors have satisfied nearly all of the critiques/questions I posed in the original review. In particular, I appreciate that the code base has been provided via Jupyter notebook. However, there appears to be a discrepancy between the Figures (specifically Figure 2a) and the notebook: Although the RNA UMAP presented in this Figure appears to be identical the projection generated in the Jupyter notebook, it appears that the two chromatin UMAPs in Figure 2a are different than what was generated in the notebook. Were the cluster/cell type definition analyses performed based on the UMAPs presented in Figure 2a, or the ones in the Jupyter notebook? The authors should ensure the analysis and projections are fully consistent with each other, lest the conclusions be altered based on the projection used. Moreover, if the cluster analyses change materially based on the projection used, this might undermine the conclusions and should be remedied/noted in some way.

Response: We express our gratitude to the reviewer for bringing up the observed discrepancy. We acknowledge that the stochastic nature of the UMAP algorithm can lead to slight variations in the generated plots unless a specific random seed state is set. When preparing a “cleaned-up” version of Jupyter Notebook, we re-run all the necessary steps without specifying the same random seed, thus produced a different UMAP. However, the Louvian clustering and cell type annotations are independent of the UMAP embedding, which serves solely as a visual representation of cell similarity. Therefore, the downstream analysis and conclusions drawn in

our manuscript remain unchanged.

In response to this concern, we have included UMAP of histone modifications in our updated analysis notebook (<https://github.com/Xieeeee/Droplet-Paired-Tag/blob/main/02.analysis/notebook/02.Clustering.ipynb>), utilizing the exact same seed that was initially selected. The resulting UMAP plots resembles the embedding shown in our Fig.2a.

Final Decision Letter:

Message 7th Jul 2023

:

Dear Professor Ren,

We are now happy to accept your revised paper "Droplet-based Single-cell Joint Profiling of Histone Modification and Transcriptome" for publication as a Brief Communication in Nature Structural & Molecular Biology. We are looking forward to how it will be received and adopted by the community.

As soon as your article is published, you can generate your shareable link by entering the DOI of your article here: ><http://authors.springernature.com/share>
. Corresponding authors will also receive an automated email with the shareable link

Your paper will be published online soon after we receive proof corrections and will appear in print in the next available issue. You can find out your date of online publication by contacting the production team shortly after sending your proof corrections. Content is published online weekly on Mondays and Thursdays, and the embargo is set at 16:00 London time (GMT)/11:00 am US Eastern time (EST) on the day of publication. Now is the time to inform your Public Relations or Press Office about your paper, as they might be interested in promoting its publication. This will allow them time to prepare an accurate and satisfactory press release. Include your manuscript tracking number (NSMB-BC47527B) and our journal name, which they will need when they contact our press office.

About one week before your paper is published online, we shall be distributing a press release to news organizations worldwide, which may very well include details of your work. We are happy for your institution or funding agency to prepare its own press release, but it must mention the embargo date and Nature Structural & Molecular Biology. If you or your Press Office have any enquiries in the meantime, please contact press@nature.com.

Please note that Nature Structural & Molecular Biology is a Transformative Journal (TJ). Authors may publish their research with us through the traditional subscription access route or make their paper immediately open access through payment of an article-processing charge (APC). Authors will not be required to make a final decision about access to their article until it has been accepted. <https://www.springernature.com/gp/open-research/transformative-journals>> Find out more about Transformative Journals

Sincerely,

Dimitris Typas
Associate Editor
Nature Structural & Molecular Biology
ORCID: 0000-0002-8737-1319
